# Turbocharging Gaussian Process Inference with Approximate Sketch-and-Project

**Pratik Rathore**
Stanford University
pratikr@stanford.edu

**Zachary Frangella**
Stanford University
zfran@stanford.edu

**Sachin Garg**
University of Michigan
sachg@umich.edu

**Shaghayegh Fazliani**
Stanford University
fazliani@stanford.edu

**Michał Dereziński**
University of Michigan
derezin@umich.edu

**Madeleine Udell**
Stanford University
udell@stanford.edu

## Abstract

Gaussian processes (GPs) play an essential role in biostatistics, scientific machine learning, and Bayesian optimization for their ability to provide probabilistic predictions and model uncertainty. However, GP inference struggles to scale to large datasets (which are common in modern applications), since it requires the solution of a linear system whose size scales quadratically with the number of samples in the dataset. We propose an approximate, distributed, accelerated sketch-and-project algorithm (`ADASAP`) for solving these linear systems, which improves scalability. We use the theory of determinantal point processes to show that the posterior mean induced by sketch-and-project rapidly converges to the true posterior mean. In particular, this yields the first efficient, condition number-free algorithm for estimating the posterior mean along the top spectral basis functions, showing that our approach is principled for GP inference. `ADASAP` outperforms state-of-the-art solvers based on conjugate gradient and coordinate descent across several benchmark datasets and a large-scale Bayesian optimization task. Moreover, `ADASAP` scales to a dataset with $> 3 \cdot 10^8$ samples, a feat which has not been accomplished in the literature.

## 1 Introduction

Gaussian processes (GPs) are a mainstay of modern machine learning and scientific computing, due to their ability to provide probabilistic predictions and handle uncertainty quantification. Indeed, GPs arise in applications spanning Bayesian optimization [Hernández-Lobato et al., 2017], genetics [McDowell et al., 2018], health care analytics [Cheng et al., 2020], materials science [Frazier and Wang, 2016], and partial differential equations [Chen et al., 2025].

GP inference for a dataset with $n$ samples requires the solution of a dense $n \times n$ linear system, which is challenging to solve at large scale. Direct methods like Cholesky decomposition require $\mathcal{O}(n^3)$ computation, limiting them to $n \sim 10^4$. Consequently, two main approaches have arisen for large-scale inference: (i) exact inference based on iterative methods for solving linear systems [Wang et al., 2019, Lin et al., 2023, 2024] and (ii) approximate inference based on inducing points and variational methods [Titsias, 2009, Hensman et al., 2013]. The state-of-the-art approaches for large-scale inference are preconditioned conjugate gradient (PCG) [Wang et al., 2019] and stochastic dual descent (SDD) [Lin et al., 2024]. PCG offers strong convergence guarantees and good performance on ill-conditioned problems, but is slow when $n \sim 10^6$. In contrast, SDD scales to $n \gg 10^6$, but lacks strong theoretical guarantees, can slow down in the face of ill-conditioning, and introduces a stepsize parameter which can be challenging to tune. Inducing points and variational methods [Titsias, 2009, Hensman et al., 2013] scale to large datasets, but are often outperformed by exact

39th Conference on Neural Information Processing Systems (NeurIPS 2025).

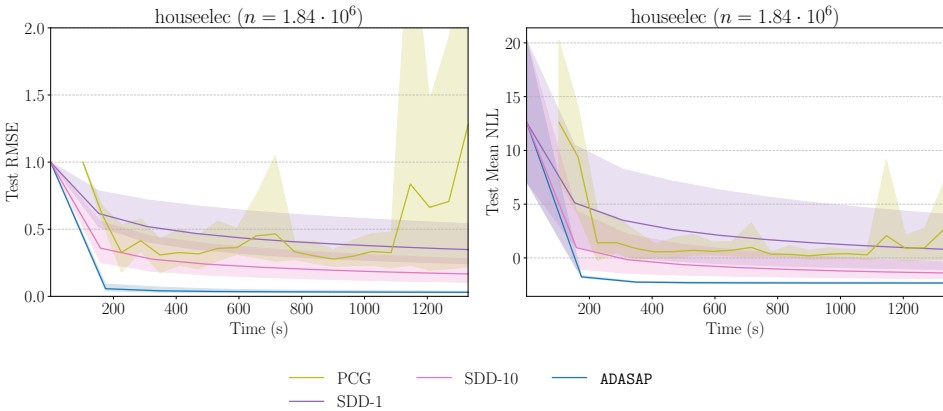

Figure 1: `ADASAP` attains lower root mean square error (RMSE) and mean negative log likelihood (NLL) than start-of-the-art methods SDD [Lin et al., 2024] and PCG on the houseelec dataset. SDD-1 and SDD-10 correspond to two particular stepsize selections for the SDD method. The solid lines indicate the mean performance of each method, while the shaded regions indicate the range between worst and best performance of each method over five random splits of the data.

inference [Wang et al., 2019, Lin et al., 2023, 2024]. Altogether, the current state of algorithms for large-scale GP inference is unsatisfactory, as practitioners have to trade-off between quality of inference, robustness to ill-conditioning, ease of setting hyperparameters, and scalability.

To address this gap in the literature, we introduce the **A**pproximate **D**istributed **A**ccelerated **S**ketch-**a**nd-**P**roject (`ADASAP`) algorithm. `ADASAP` is rooted in the sketch-and-project framework of Gower and Richtárik [2015], and obtains the robustness to ill-conditioning of PCG and the scalability of SDD, while having reliable, default hyperparameters that work well in practice. For example, `ADASAP` dramatically outperforms tuned SDD and PCG on a dataset with $n > 10^6$ samples (Fig. 1).

Our contributions are as follows:

1. We develop `ADASAP` for large-scale GP inference. `ADASAP` uses approximate preconditioning and acceleration to address ill-conditioning, and uses distributed computing to improve the speed of bottleneck operations. `ADASAP` also comes with effective default hyperparameters that work out of the box.

2. Using the theory of determinantal point processes, we show that sketch-and-project-style methods converge faster than stochastic first-order methods like SDD in the presence of ill-conditioning. In particular, we give a first-of-its-kind condition number-free time complexity bound for estimating the GP posterior mean to moderate accuracy, explaining the excellent test error performance of `ADASAP`.

3. We empirically verify that `ADASAP`, with its default hyperparameters, outperforms state-of-the-art competitors on benchmark large-scale GP inference tasks, and is capable of scaling to a dataset with $n > 3 \cdot 10^8$ samples.

4. We show `ADASAP` yields the best performance on a large-scale Bayesian optimization task from Lin et al. [2023].

## 2 GP Regression and Inference

Let $\mathcal{X}$ be a set. A random function $f \colon \mathcal{X} \to \mathbb{R}$ is called a *Gaussian process* if for any finite subset $\{x_1, x_2, \ldots, x_n\} \subset \mathcal{X}$ of points (where $n$ is any positive integer), the random vector $(f(x_1), f(x_2), \ldots, f(x_n))$ follows a multivariate Gaussian distribution [Rasmussen and Williams, 2005]. A Gaussian process is typically denoted as $f \sim \mathcal{GP}(m, k)$, where $m \colon \mathcal{X} \to \mathbb{R}$ is the mean function and $k \colon \mathcal{X} \times \mathcal{X} \to \mathbb{R}$ is the covariance function (or kernel). Throughout this paper, the kernel function $k$ follows broadcasting conventions, where operations between individual points and sets of points produce vectors or matrices of kernel evaluations as appropriate.

Suppose we have a Gaussian process prior $f \sim \mathcal{GP}(m, k)$. Given observations $(X, y) \in \mathbb{R}^{n \times d} \times \mathbb{R}^n$ and likelihood variance $\lambda > 0$, we would like to perform inference using the GP posterior, i.e., (i) sampling from the posterior and (ii) computing the posterior mean.

For conciseness, define $K := k(X, X) \in \mathbb{R}^{n \times n}$. Then a sample from the GP posterior is a random function $f_n \sim \mathcal{GP}(m_n, k_n)$ [Rasmussen and Williams, 2005], where

$$m_n(\cdot) = m(\cdot) + k(\cdot, X)(K + \lambda I)^{-1} y, \tag{1}$$

$$k_n(\cdot, \cdot) = k(\cdot, \cdot) - k(\cdot, X)(K + \lambda I)^{-1} k(X, \cdot). \tag{2}$$

The standard approach for drawing posterior samples at locations $X_*$ is to perform a linear transformation of standard Gaussian random variables $\zeta \sim \mathcal{N}(0, I)$ [Wilson et al., 2020, 2021]:

$$f_n(X_*) = m_n(X_*) + k_n(X_*, X_*)^{1/2} \zeta. \tag{3}$$

Unfortunately, the approach in (3) does not scale when $|X_*|$ is large, since the matrix square root $k_n(X_*, X_*)^{1/2}$ requires $O(|X_*|^3)$ computation.

## 2.1 Pathwise conditioning

To address the scaling challenges in (3), Wilson et al. [2020, 2021] develop pathwise conditioning. Pathwise conditioning rewrites (3) as

$$f_n(X_*) = f(X_*) + k_n(X_*, X)(K + \lambda I)^{-1}(y - f(X_n) - \zeta), \quad \zeta \sim \mathcal{N}(0, \lambda I), \quad f \sim \mathcal{GP}(0, k). \tag{4}$$

Equation (4) enables scalable sampling from the posterior as $f$ can be approximated using random features [Rahimi and Recht, 2007, Rudi and Rosasco, 2017] for $k$, which costs $\mathcal{O}(q|X_*|)$, where $q$ is the number of random features. This yields a significant reduction in cost from the standard posterior sampling scheme in (3): by leveraging pathwise conditioning, posterior sampling can be performed by generating $f$ via random features [Wilson et al., 2020, 2021] and solving $(K + \lambda I)^{-1}(y - f(X_n) - \zeta)$ using an iterative method like PCG or SDD. Our experiments (Section 5) use pathwise conditioning to sample from the posterior.

## 2.2 Drawbacks of state-of-the-art exact inference methods

The current state-of-the-art for large-scale posterior sampling is SDD [Lin et al., 2024], a stochastic first-order algorithm combining coordinate descent [Richtárik and Takáč, 2014, Qu and Richtárik, 2016] with momentum and geometric averaging. Stochastic methods like SDD can better handle single (or lower) precision than PCG [Wu et al., 2024, Rathore et al., 2025], which is essential to maximizing GPU performance and reducing storage costs [Abdelfattah et al., 2021]. While SDD overcomes the scalability challenges associated with PCG, it faces challenges due to the ill-conditioning of $K + \lambda I$, which determines its worst-case convergence rate [Nemirovski and Yudin, 1983]. Therefore, reliable large-scale inference requires a method that has cheaper per-iteration costs than PCG and can handle the ill-conditioning of $K + \lambda I$. However, to the best of our knowledge, no existing method effectively tackles both issues: SGD and SDD offer speed and scalability but suffer from ill-conditioning, while PCG offers robustness to ill-conditioning but struggles to scale beyond $n > 10^6$. In Section 3 we propose SAP, which is robust to ill-conditioning; in Section 4, we develop ADASAP, an extension of SAP that scales to large datasets.

# 3 Sketch-and-project for GP Inference

Section 2 shows the main challenge of large-scale GP inference is the need to solve a large linear system with the ill-conditioned matrix $K + \lambda I$, which cannot be done satisfactorily with PCG or SDD. We introduce the SAP algorithm, which effectively addresses the conditioning challenges of GP inference. However, SAP alone does not address the scalability challenges of GP inference, motivating the development of ADASAP in Section 4.

## 3.1 Sketch-and-project

We formally present SAP in Algorithm 1. SAP resembles randomized block coordinate descent, except that it employs *subspace preconditioning*. That is, it preconditions the subspace gradient

**Algorithm 1** SAP for $K_\lambda W = Y$

---

**Require:** blocksize $b$, distribution of row indices $\mathcal{D}$ over $[n]^b$, number of iterations $T_{\max}$, initialization $W_0 \in \mathbb{R}^{n \times n_{\text{rhs}}}$, averaging boolean tail_average

  **for** $t = 0, 1, \dots, T_{\max} - 1$ **do**
    Sample row indices $\mathcal{B}$ of size $b$ from $[n]$ according to $\mathcal{D}$
    $D_t \leftarrow I_\mathcal{B}^T (K_{\mathcal{B}\mathcal{B}} + \lambda I)^{-1}(K_{\mathcal{B}n}W_t + \lambda I_\mathcal{B}W_t - Y_\mathcal{B})$   ▷ Search direction; costs $\mathcal{O}(bnn_{\text{rhs}} + b^3)$
    $W_{t+1} \leftarrow W_t - D_t$                                   ▷ Update parameters; costs $\mathcal{O}(bn_{\text{rhs}})$
    **if** tail_average **then**
      $\bar{W}_{t+1} \leftarrow 2/(t+1) \sum_{j=(t+1)/2}^{t} W_j$
    **end if**
  **end for**
  **return** $\bar{W}_T$ if tail_average else $W_T$           ▷ Approximate solution to $K_\lambda W = Y$

---

$K_{\mathcal{B}n}W_t + \lambda I_\mathcal{B}W_t - Y_\mathcal{B}$ by the inverse of the subspace Hessian $K_{\mathcal{B}\mathcal{B}} + \lambda I$. Thus, unlike SDD, SAP includes second-order information in its search direction. In addition, we have also augmented vanilla SAP from Gower and Richtárik [2015] with tail averaging [Jain et al., 2018, Epperly et al., 2024a], which helps address the inherent noise in the iterates, and plays a crucial role in our analysis.

SAP (assuming the number of right-hand sides $n_{\text{rhs}} = 1$ for simplicity) only costs $\mathcal{O}(bn + b^3)$ per-iteration, which is significantly smaller than the $\mathcal{O}(n^2)$ iteration cost of PCG, while being comparable with SDD, which costs $\mathcal{O}(bn)$ per-iteration. Furthermore, PCG typically requires memory linear in $n$ to store the preconditioner, while SAP only incurs and additional $\mathcal{O}(b^2)$ storage cost. Thus, SAP strikes a balance between PCG and methods like SDD. It incorporates second-order information, like PCG, but does so at a modest increase in cost relative to SDD and SGD. By leveraging the theory of determinantal point processes [Kulesza and Taskar, 2012], we show that tail-averaged SAP improves dependence upon the condition number in the early phase of the convergence, which establishes it as a potential solution to the conditioning challenges discussed in Section 2, particularly in the presence noisy data and generalization error.

## 3.2 Fast convergence along top-$\ell$ subspace for GP inference

Prior works [Dicker et al., 2017, Lin et al., 2020, 2023] have shown that the components of the solution most relevant for learning lie along the dominant eigenvectors of $K$, which correspond to the dominant spectral basis functions of the reproducing kernel Hilbert space (RKHS) $\mathcal{H}$ induced by kernel $k$ with respect to $X$. Lin et al. [2023] shows that SGD enjoys an improved early convergence along these directions, but attains a slow sublinear rate asymptotically. Motivated by these observations, we study the convergence rate of tail-averaged SAP along the dominant spectral basis functions, showing that it obtains a two-phase convergence rate: an initial condition number-free sublinear rate, followed by an asymptotic linear rate where the condition number dependence is mitigated by the blocksize $b$. The proofs of the results in this section appear in Appendix B.3.

Following Lin et al. [2023], we define the spectral basis functions of $\mathcal{H}$ as

$$u^{(i)} = \underset{u \in \mathcal{H}, \|u\|_\mathcal{H}=1}{\operatorname{argmax}} \left\{ \|u(X)\|^2 \mid \langle u, u^{(j)} \rangle_\mathcal{H} = 0 \text{ for } j < i \right\}.$$

The spectral basis functions $u^{(1)}, u^{(2)}, \dots$ are defined in such a way that each of them takes maximal values at the observation points while being orthogonal to the previous ones. Hence, to accurately estimate the posterior mean in the data-dense regions, it is most important to minimize the error along the dominant basis functions.

We focus on posterior mean estimation, which corresponds to solving the linear system $K_\lambda W = y$ using tail-averaged SAP. Recall that, given a sequence of weight vectors $W_1, \dots, W_t$ produced by sketch-and-project (which are vectors, not matrices, as $n_{\text{rhs}} = 1$), tail averaging obtains a weight estimate by averaging the second half of this sequence. We use these weights to define an estimate of the GP posterior mean $m_n$, which is given by $\hat{m}_t = k(\cdot, X)\bar{W}_t$.

Theorem 3.2 is our main theoretical result. It provides a convergence guarantee for the SAP posterior mean estimate $\hat{m}_t$ along the top-$\ell$ spectral basis functions, where $\ell$ is a parameter that can be chosen

freely. Theorem 3.2 shows that while the estimate converges to the true posterior mean over the entire RKHS $\mathcal{H}$, its initial convergence rate along the dominant basis functions is even faster. Our theoretical results rely on the *smoothed condition number*, which we define below.

**Definition 3.1** (Smoothed condition number). Let $A \in \mathbb{R}^{n \times n}$ be a positive-semidefinite matrix with eigenvalues $\lambda_1 \geq \ldots \geq \lambda_n$. We define the smoothed condition number $\phi(b, p)$ of $A$ as

$$\phi(b, p) \coloneqq \frac{1}{b} \sum_{i > b} \frac{\lambda_i}{\lambda_p}.$$

The smoothed condition number controls both the SGD-style sublinear convergence rate and the PCG-style linear convergence rate in Theorem 3.2. Note that, for notational convenience, we use $2b$ to denote the SAP blocksize in the below statement.

**Theorem 3.2.** *Suppose we have a kernel matrix $K \in \mathbb{R}^{n \times n}$, observations $y \in \mathbb{R}^n$ from a GP prior with likelihood variance $\lambda \geq 0$, and let $\phi(\cdot, \cdot)$ denote the smoothed condition number of $K + \lambda I$. Given any $\ell \in [n]$, let $\mathrm{proj}_\ell$ denote orthogonal projection onto the span of $u^{(1)}, ..., u^{(\ell)}$. For any $1 \leq b \leq n/2$, in $\tilde{\mathcal{O}}(nb^2)$ time we can construct a distribution over row index subsets $\mathcal{B}$ of size $2b$ such that SAP (Algorithm 1) initialized at zero after $t$ iterations satisfies*

$$\mathbb{E} \left\| \mathrm{proj}_\ell(\hat{m}_t) - \mathrm{proj}_\ell(m_n) \right\|_{\mathcal{H}}^2 \leq \min \left\{ \frac{8\phi(b, \ell)}{t}, \left( 1 - \frac{1}{2\phi(b, n)} \right)^{t/2} \right\} \|y\|_{K_\lambda^{-1}}^2.$$

This result provides a composite convergence rate for SAP: the first rate, $\frac{8\phi(b,\ell)}{t}$, is a sublinear rate, similar to SGD [Lin et al., 2023], while the second rate, $\left( 1 - \frac{1}{2\phi(b,n)} \right)^{t/2}$, is a linear convergence rate. The linear rate is independent of the subspace norm, and thus, in particular, also applies for $\ell = n$, i.e., when comparing posterior means within the entire space. However, for $\ell \ll n$, the sublinear rate wins out in the early phase of the convergence, since $\phi(b, \ell) \ll \phi(b, n)$.

For ill-conditioned kernels with small prior variance $\lambda$, the linear rate $1 - 1/(2\phi(b, n))$ is close to 1 (as $b$ increases, the preconditioning effect becomes stronger, which improves the rate). On the other hand, the constant in the sublinear rate $\phi(b, \ell)$ is much smaller, since it takes advantage of the top-$\ell$ subspace norm, which makes it effectively independent of the conditioning of $K$.

If the SAP blocksize is proportional to $\ell$ and $K + \lambda I$ exhibits *any* polynomial spectral decay, then $\phi(b, \ell) = \mathcal{O}(1)$, making the sublinear rate condition number-free. Most popular kernels exhibit polynomial spectral decay [Ma and Belkin, 2017, Kanagawa et al., 2018]. Furthermore, polynomial spectral decay is a standard assumption in the generalization analysis of kernel methods in the fixed and random design settings [Caponnetto and DeVito, 2007, Bach, 2013, Rudi et al., 2015]. To the best of our knowledge, our result is the first of its kind, as previous comparable guarantees for stochastic methods [Lin et al., 2023, Proposition 1] still depend on the condition number. We note that our condition number-free rate is attained only when seeking moderately accurate estimates of the posterior mean (which is often the case due to the presence of noise in the data). When seeking highly accurate estimates, the asymptotic (condition number dependent) linear rate will eventually take over. We illustrate these claims with the following corollary:

**Corollary 3.3.** *Suppose that the matrix $K$ exhibits polynomial spectral decay, i.e., $\lambda_i(K) = \Theta(i^{-\beta})$ for some $\beta > 1$. Then for any $\ell \in \{1, \ldots, n\}$ and $\epsilon \in (0, 1)$, choosing block size $b = 4\ell$ we can find $\hat{m}$ that with probability at least $0.99$ satisfies $\|\mathrm{proj}_\ell(\hat{m}) - \mathrm{proj}_\ell(m_n)\|_{\mathcal{H}}^2 \leq \epsilon \cdot \|\mathrm{proj}_\ell(m_n)\|_{\mathcal{H}}^2$ in*

$$\tilde{\mathcal{O}} \left( (n^2 + n\ell^2) \min \left\{ \frac{1}{\epsilon}, \left( 1 + \frac{\ell(\lambda_\ell(K) + \lambda)}{n(\lambda_n(K) + \lambda)} \right) \log \left( \frac{1}{\epsilon} \right) \right\} \right) \quad \text{time.}$$

*Remark* 3.4. The $\tilde{\mathcal{O}}((n^2 + n\ell^2)/\epsilon)$ runtime (phase one, attained for moderate accuracy $\epsilon$) is entirely independent of the condition number of $K$ and the likelihood variance $\lambda$. In Appendix B, we show that when $\lambda$ is sufficiently small (the highly ill-conditioned case), then the phase one complexity of sketch-and-project can be further improved by increasing the block size $b$ to attain $\tilde{\mathcal{O}}(nb^2 + (n^2 + nb^2)(\ell/b)^{\beta-1}/\epsilon)$ runtime. In particular, when $\ell \ll b \ll \sqrt{n}$, we obtain meaningful convergence in $o(n^2)$ time, i.e., before reading all of the data. Thus, the top-$\ell$ subspace convergence of SAP can actually benefit from ill-conditioned kernels as the leading spectral basis functions become more dominant.

Although our theoretical results require tail averaging, we do not believe it is needed for good practical performance of sketch-and-project algorithms. Indeed, we run sketch-and-project with and without tail averaging in Appendix C.6, and we find that (i) tail averaging does not improve subspace convergence by a substantial amount and (ii) tail averaging performs worse than not using tail averaging at larger blocksizes.

## 4  `ADASAP`: Approximate, Distributed, Accelerated `SAP`

We introduce `ADASAP` (Algorithm 2), a scalable extension of `SAP` for GP inference. We first introduce the modifications made to `SAP` for scalability (Section 4.1), before presenting `ADASAP` in full (Section 4.2). For a detailed discussion of how `ADASAP` relates to prior work, please see Appendix A.

### 4.1  The key ingredients: approximation, distribution, and acceleration

We begin by discussing the essential elements of the `ADASAP` algorithm: (i) approximate subspace preconditioning, (ii) distribution, (iii) acceleration, and how they enhance performance.

#### 4.1.1  Approximate subspace preconditioning

`SAP` enjoys significant improvements over PCG, as it only requires $\mathcal{O}(bn + b^3)$ computation per-iteration. Unfortunately, `SAP` is limited in how large a blocksize $b$ it may use, as factoring $K_{\mathcal{BB}} + \lambda I$ to perform subspace preconditioning costs $\mathcal{O}(b^3)$. This is problematic, as Theorem 3.2 shows the convergence rate of `SAP` improves as $b$ increases.

To address this challenge, `ADASAP` draws inspiration from previous works [Erdogdu and Montanari, 2015, Frangella et al., 2024b, Rathore et al., 2025], which replace exact linear system solves in iterative algorithms with inexact solves based on low-rank approximations. `ADASAP` replaces $K_{\mathcal{BB}}$ in `SAP` with a rank-$r$ randomized Nyström approximation $\hat{K}_{\mathcal{BB}}$ [Williams and Seeger, 2000, Gittens and Mahoney, 2016, Tropp et al., 2017]. This strategy is natural, as kernels exhibit approximate low-rank structure [Bach, 2013, Rudi et al., 2015, Belkin, 2018]. Indeed, Rathore et al. [2025] develops an approximate SAP solver for kernel ridge regression with one right-hand side, based on approximating $K_{\mathcal{BB}}$ by $\hat{K}_{\mathcal{BB}}$, and observes strong empirical performance.

`ADASAP` computes $\hat{K}_{\mathcal{BB}}$ following the numerically stable procedure from Tropp et al. [2017], which is presented in Appendix C. The key benefit is that $(\hat{K}_{\mathcal{BB}} + \rho I)^{-1}$ can be applied to vectors in $\mathcal{O}(br)$ time (Appendix C), where $\rho > 0$ is a damping parameter to ensure invertibility. This reduces the cost compared to the $\mathcal{O}(b^3)$ exact SAP update and allows `ADASAP` to use larger blocksizes: on the taxi dataset ($n = 3.31 \cdot 10^8$), `ADASAP` uses blocksize $b = 1.65 \cdot 10^5$.

#### 4.1.2  Distributed matrix-matrix products

`SAP` with approximate subspace preconditioning allows large blocksize $b$, but two bottlenecks remain: computing $K_{\mathcal{B}n} W_t$ and constructing the sketch $K_{\mathcal{BB}} \Omega$. The former costs $\mathcal{O}(bn)$, while the latter costs $\mathcal{O}(b^2 r)$. As matrix-matrix multiplication is embarrassingly parallel, distributed multi-GPU acceleration can address these bottlenecks. By partitioning $K_{\mathcal{B}n}$ across $\mathrm{N}_{\mathrm{workers}}$ GPUs, `ColDistMatMat` significantly reduces the time to compute the product $K_{\mathcal{B}n} \Omega$. Similarly, `RowDistMatMat` for $K_{\mathcal{BB}} \Omega$ reduces the time to compute the sketch. On our largest dataset, taxi, using 4 GPUs achieves a $3.4\times$ speedup (Fig. 4), which corresponds to near-perfect parallelism.

#### 4.1.3  Nesterov acceleration

Prior work has shown Nesterov acceleration [Nesterov, 1983] improves the convergence rate of `SAP` and approximate SAP [Tu et al., 2017, Gower et al., 2018, Dereziński et al., 2024, Rathore et al., 2025]. Hence we use Nesterov acceleration in `ADASAP` to improve convergence.

### 4.2  `ADASAP` algorithm

The pseudocode for `ADASAP` is presented in Algorithm 2. `ADASAP` uses tail averaging, approximate subspace preconditioning, distributed matrix-matrix products, and Nesterov acceleration. Assuming

---

**Algorithm 2** `ADASAP` for $K_\lambda W = Y$

---

**Require:** distribution of row indices $\mathcal{D}$ over $[n]^b$, distribution over Nyström sketching matrices $\mathcal{D}_{\text{Nyström}}$ over $\mathbb{R}^{b \times r}$, number of iterations $T_{\max}$, initialization $W_0 \in \mathbb{R}^{n \times n_{\text{rhs}}}$, workers $\{\mathcal{W}_1, \dots, \mathcal{W}_{\text{N}_{\text{workers}}}\}$, averaging boolean tail_average, acceleration parameters $\mu, \nu$

   **# Initialize acceleration parameters**
   $\beta \leftarrow 1 - \sqrt{\mu/\nu}, \quad \gamma \leftarrow 1/\sqrt{\mu\nu}, \quad \alpha \leftarrow 1/(1 + \gamma\nu)$
   $V_0 \leftarrow W_0, Z_0 \leftarrow W_0$

   **for** $t = 0, 1, \dots, T_{\max} - 1$ **do**
      **# Step I: Compute matrix-matrix product** $K_{\mathcal{B}n}W_t$        $\triangleright$ Costs $\mathcal{O}(bnn_{\text{rhs}}/\text{N}_{\text{workers}})$
      Sample row indices $\mathcal{B}$ of size $b$ from $[n]$ according to $\mathcal{D}$
      $K_{\mathcal{B}n}W_t \leftarrow \text{ColDistMatMat}(W_t, \mathcal{B}, \{\mathcal{W}_1, \dots, \mathcal{W}_{\text{N}_{\text{workers}}}\})$        $\triangleright$ Algorithm 3

      **# Step II: Compute Nyström sketch** $K_{\mathcal{B}\mathcal{B}}\Omega$        $\triangleright$ Costs $\mathcal{O}(rb^2/\text{N}_{\text{workers}})$
      Sample $\Omega \in \mathbb{R}^{b \times n}$ from $\mathcal{D}_{\text{Nyström}}$
      $K_{\mathcal{B}\mathcal{B}}\Omega \leftarrow \text{RowDistMatMat}(\Omega, \mathcal{B}, \{\mathcal{W}_1, \dots, \mathcal{W}_{\text{N}_{\text{workers}}}\})$        $\triangleright$ Algorithm 4

      **# Step III: Compute Nyström approximation and get stepsize**        $\triangleright$ Costs $\mathcal{O}(b^2 + br^2)$
      $U, S \leftarrow \text{RandNysAppx}(K_{\mathcal{B}\mathcal{B}}\Omega, \Omega, r)$        $\triangleright$ Algorithm 5
      $P_{\mathcal{B}} \leftarrow USU^T + (S_{rr} + \lambda)I$        $\triangleright$ Get preconditioner. Never explicitly formed!
      $\eta_{\mathcal{B}} \leftarrow \text{GetStepsize}(P_{\mathcal{B}}, K_{\mathcal{B}\mathcal{B}} + \lambda I)$        $\triangleright$ Algorithm 6

      **# Step IV: Compute updates using acceleration**        $\triangleright$ Costs $\mathcal{O}(brn_{\text{rhs}} + nn_{\text{rhs}})$
      $D_t \leftarrow I_{\mathcal{B}}^T P_{\mathcal{B}}^{-1}(K_{\mathcal{B}n}W_t + \lambda I_{\mathcal{B}}W_t - Y_{\mathcal{B}})$
      $W_{t+1}, V_{t+1}, Z_{t+1} \leftarrow \text{NestAcc}(W_t, V_t, Z_t, D_t, \eta_{\mathcal{B}}, \beta, \gamma, \alpha)$        $\triangleright$ Algorithm 7

      **if** tail_average **then**
         $\bar{W}_{t+1} \leftarrow 2/(t+1) \sum_{j=(t+1)/2}^{t} W_j$
      **end if**
   **end for**
   **return** $\bar{W}_T$ if tail_average else $W_T$        $\triangleright$ Approximate solution to $K_\lambda W = Y$

---

perfect parallelism, `ADASAP` has per iteration runtime of $\mathcal{O}(nn_{\text{rhs}}b/\text{N}_{\text{workers}} + b^2r/\text{N}_{\text{workers}})$, a significant improvment over the $\mathcal{O}(nn_{\text{rhs}}b + b^3)$ iteration time of `SAP`. The per-iteration runtime of `ADASAP` is comparable to distributed SDD—in other words, `ADASAP` effectively preconditions the problem, reducing the total iterations required, without significantly slowing down each iteration. Moreover, unlike SDD, `ADASAP` automatically sets the stepsize at each iteration, removing the need for expensive tuning. By default, we set the acceleration parameters $\mu$ and $\nu$ to $\lambda$ and $n/b$, respectively (as done in Rathore et al. [2025]), and find they yield excellent performance in Section 5.

### 4.2.1 Theory vs. Practice

Our theory in Section 3 does not cover the approximate preconditioning, acceleration, or uniform sampling that is used in `ADASAP`. Despite the theory-practice gap between `SAP` (Algorithm 1) and `ADASAP` (Algorithm 2), we believe the theory developed for `SAP` can be extended to `ADASAP`. This belief is rooted in Dereziński and Yang [2024], Dereziński et al. [2024], Rathore et al. [2025], which show that `SAP` methods still converge when using approximate preconditioning, acceleration, and uniform sampling. We leave this extension as a direction for future research.

For simplicity, our experiments run `ADASAP` without tail averaging. Tail averaging is needed to establish Theorem 3.2, but we do not expect it to yield significant practical improvements (Appendix C.6). This is in line with the SGD literature, where averaging leads to better theoretical convergence rates, but the last iterate delivers similar performance in practice [Shamir and Zhang, 2013, Johnson and Zhang, 2013].

Table 1: RMSE and mean negative log-likelihood (NLL) obtained by `ADASAP` and competitors on the test set. The results are averaged over five 90%-10% train-test splits of each dataset. We **bold** a result if it gets to within $0.01$ of the best found RMSE or mean NLL.

| | Dataset | yolanda | song | benzene | malonaldehyde | acsincome | houseelec |
|---|---|---|---|---|---|---|---|
| | $n$ | $3.60 \cdot 10^5$ | $4.64 \cdot 10^5$ | $5.65 \cdot 10^5$ | $8.94 \cdot 10^5$ | $1.50 \cdot 10^6$ | $1.84 \cdot 10^6$ |
| | $d$ | 100 | 90 | 66 | 36 | 9 | 9 |
| | $k$ | RBF | Matérn-3/2 | Matérn-5/2 | Matérn-5/2 | RBF | Matérn-3/2 |
| Test RMSE | `ADASAP` | **0.795** | **0.752** | **0.012** | **0.015** | **0.789** | **0.027** |
| | `ADASAP-I` | 0.808 | 0.782 | 0.168 | 0.231 | **0.795** | 0.066 |
| | SDD-1 | 0.833 | 0.808 | 0.265 | 0.270 | 0.801 | 0.268 |
| | SDD-10 | **0.801** | 0.767 | 0.112 | Diverged | **0.792** | 0.119 |
| | SDD-100 | Diverged | Diverged | Diverged | Diverged | Diverged | Diverged |
| | PCG | **0.795** | **0.752** | 0.141 | 0.273 | 0.875 | 1.278 |
| Test Mean NLL | `ADASAP` | **1.179** | **1.121** | **-2.673** | **-2.259** | **1.229** | **-2.346** |
| | `ADASAP-I` | 1.196 | 1.170 | -0.217 | 0.466 | **1.235** | -2.185 |
| | SDD-1 | 1.225 | 1.203 | 0.531 | 0.903 | 1.242 | -0.281 |
| | SDD-10 | **1.187** | 1.149 | -0.762 | Diverged | **1.232** | -1.804 |
| | SDD-100 | Diverged | Diverged | Diverged | Diverged | Diverged | Diverged |
| | PCG | **1.179** | **1.121** | -0.124 | 0.925 | 1.316 | 2.674 |

## 5 Experiments

We present three sets of experiments showing `ADASAP` outperforms state-of-the-art methods for GP inference: (i) GP inference on large benchmark datasets (Section 5.1), (ii) GP inference on huge-scale transportation data analysis with $n > 3 \cdot 10^8$ samples (Section 5.2), and (iii) the Bayesian optimization task from Lin et al. [2023] (Section 5.3). We evaluate `ADASAP` against the following competitors:

- `ADASAP-I`: A variant of `ADASAP` where the subspace preconditioner $P_\mathcal{B}$ is set to the identity matrix. `ADASAP-I` is the same as accelerated randomized block coordinate descent.

- SDD: The coordinate descent method introduced by Lin et al. [2024]. We tune SDD using three different stepsizes, and denote these variants by SDD-1, SDD-10, and SDD-100.

- PCG: A combination of block CG [O'Leary, 1980] with Nyström preconditioning [Frangella et al., 2023].

Our experiments are run in single precision on 48 GB NVIDIA RTX A6000 GPUs using Python 3.10, PyTorch 2.6.0 [Paszke et al., 2019], and CUDA 12.5. We use 2, 3, and 1 GPU(s) per experiment in Sections 5.1 to 5.3, respectively. Code for reproducing our experiments is available at https://github.com/pratikrathore8/scalable_gp_inference.

Additional details are in Appendix D.

### 5.1 GP inference on large-scale datasets

We benchmark on six large-scale regression datasets from the UCI repository, OpenML, and sGDML [Chmiela et al., 2017]. The results are reported in Table 1. `ADASAP` achieves the lowest RMSE and mean negative log-likelihood (NLL) on each dataset; the NLL is computed using 64 posterior samples (via pathwise conditioning). SDD-10 is competitive with `ADASAP` on yolanda and acsincome, but performs much worse on the other datasets. SDD is also sensitive to the stepsize: SDD-1 attains a larger RMSE and NLL than `ADASAP` on all datasets, while SDD-100 diverges on all datasets. PCG obtains the same RMSE and NLL as `ADASAP` on yolanda and song, but its performance degrades for larger datasets. Additionally, Fig. 2 shows `ADASAP` achieves the lowest RMSE and NLL throughout the optimization process, demonstrating its efficiency with respect to wall-clock time.

### 5.2 Showcase: Transportation data analysis with $> 3 \cdot 10^8$ samples

To demonstrate the power of `ADASAP` on huge-scale problems, we perform GP inference on a subset of the taxi dataset (https://github.com/toddwschneider/nyc-taxi-data) with $n = 3.31 \cdot 10^8$ samples

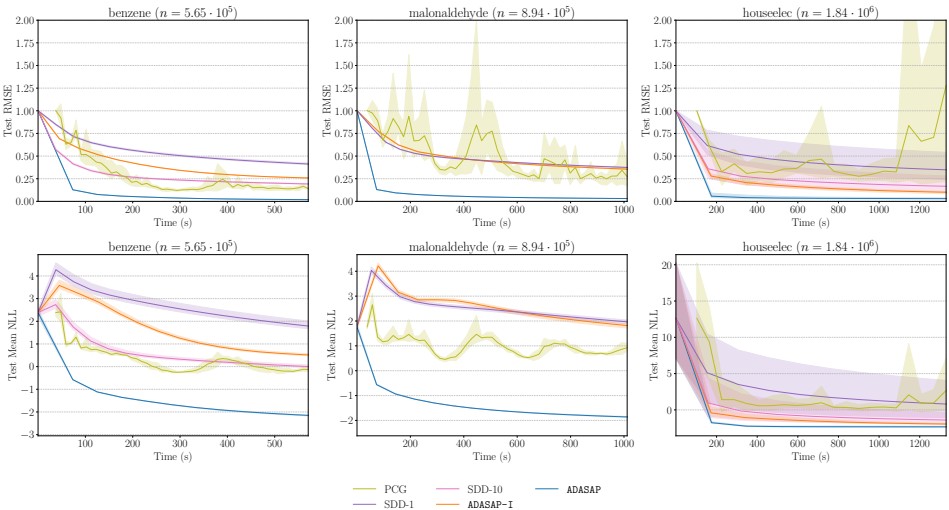

Figure 2: Performance of `ADASAP` and competitors on RMSE and mean NLL, as a function of time, for benzene, malonaldehyde, and houseelec. The solid curve indicates mean performance over random splits of the data; the shaded regions indicate the range between the worst and best performance over random splits of the data. `ADASAP` outperforms the competition.

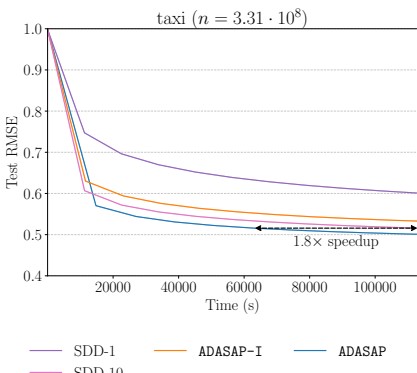

Figure 3: Comparison between `ADASAP` and competitors on transportation data analysis. `ADASAP` attains the lowest RMSE and it obtains a $1.8\times$ speed up over the second-best method, SDD-10. SDD-100 diverges and PCG runs out of memory, so they do not appear in the figure.

and dimension $d = 9$: the task is to predict taxi ride durations in New York City. To the best of our knowledge, this is the first time that full GP inference has been scaled to a dataset of this size. Due to memory constraints, we are unable to compute 64 posterior samples as done in Section 5.1 for computing NLL, so we only report RMSE.

The results are shown in Fig. 3. `ADASAP` obtains the lowest RMSE out of all the methods. Once again, SDD is sensitive to the stepsize: SDD-1 obtains an RMSE of 0.60, as opposed to `ADASAP`, which obtains an RMSE of 0.50, while SDD-100 diverges. SDD-10 obtains an RMSE of 0.52, which is similar to that of `ADASAP`. However, `ADASAP` reaches the RMSE attained by SDD-10 in 45% less time than SDD-10, which translates to a difference of 14 hours of runtime! Furthermore, PCG runs out of memory, as the memory required for storing the sketch in single precision is $3.31 \cdot 10^8 \cdot 100 \cdot 4 \approx 130$ GB, which exceeds the 48 GB of memory in the A6000 GPUs used in our experiments.

## 5.3 Large-scale Bayesian optimization

We run `ADASAP` and competitors on a variant of the synthetic large-scale Bayesian optimization tasks from Lin et al. [2023]. These Bayesian optimization tasks consist of finding the maximum of black

box functions $f : [0, 1]^8 \to \mathbb{R}$ sampled from a Matérn-3/2 Gaussian process. We use two different lengthscales (2.0 and 3.0) and 5 random functions per lengthscale. To avoid misspecification, we set the kernel of each model to match that of the black box function.

The results are shown in Table 2. ADASAP makes the biggest improvement over the random search baseline. As we have seen in the other experiments, SDD is sensitive to the stepsize: SDD-10 and SDD-100 make no progress. PCG also makes less progress than ADASAP across both lengthscales.

Table 2: Percentage improvement over random search for Bayesian optimization, averaged over five seeds. SDD-10 and SDD-100 provide no improvement over random search because they are unstable.

|  |  | Lengthscale = 2.0 | Lengthscale = 3.0 |
|---|---|---|---|
| | ADASAP | **10.42** | **13.86** |
| | ADASAP-I | 7.04 | 11.27 |
| Improv. (%) | SDD-1 | 6.50 | 11.17 |
| | SDD-10 | 0.00 | 0.00 |
| | SDD-100 | 0.00 | 0.00 |
| | PCG | 0.13 | 5.54 |

## 6 Conclusion

We introduce ADASAP, an approximate, distributed, accelerated sketch-and-project method for GP inference. We demonstrate that ADASAP outperforms state-of-the-art GP inference methods like PCG and SDD on large-scale benchmark datasets, a huge-scale dataset with $> 3 \cdot 10^8$ samples, and large-scale Bayesian optimization. Moreover, we show that SAP-style methods are theoretically principled for GP inference—we prove that SAP is the first efficient, condition number-free algorithm for estimating the posterior mean along the top spectral basis functions. Future work should extend the theoretical results for SAP to ADASAP and investigate ADASAP in lower precision (e.g., float16).

## Acknowledgments and Disclosure of Funding

We would like to thank Jihao Andreas Lin for helpful discussions regarding this work. PR, ZF, SF, and MU gratefully acknowledge support from the National Science Foundation (NSF) Award IIS-2233762, the Office of Naval Research (ONR) Awards N000142212825, N000142412306, and N000142312203, the Alfred P. Sloan Foundation, and from IBM Research as a founding member of Stanford Institute for Human-centered Artificial Intelligence (HAI). MD and SG gratefully acknowledge support from NSF Award CCF-2338655.

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

# A   Related Work

We review the literature on Gaussian process (GP) inference and sketch-and-project, which are the two key areas that serve as foundations for `ADASAP`.

## A.1   GP inference

Naive GP inference based on solving the linear systems in (1) exactly and generating posterior samples via (3) has cubic cost in number of training and test points, making it prohibitively expensive for datasets whose size exceeds $n \sim 10^4$. Given its prominence in scientific computing and machine learning, much work has been to done to scale GP inference to the big-data setting. Prior work in this area can be (roughly) divided into four approaches: (i) PCG-based inference, (ii) SGD-based inference, (iii) low-dimensional inference, and (iv) variational inference.

### A.1.1   PCG-based inference

A natural approach to scaling GP inference is to replace the exact linear system solves by a direct method with approximate linear system solves by an iterative method. Gardner et al. [2018] showed that by utilizing batched PCG and GPU acceleration, GP inference could be scaled to datasets of size $n \sim 10^5$. Wang et al. [2019] further showed that by distributing the kernel matrix across multiple GPUs, that inference could be applied to datasets of size $n \sim 10^6$. Unfortunately, these approaches still have shortcomings. While Gardner et al. [2018], Wang et al. [2019] show that exact GPs can be scaled to datasets with $n \sim 10^6$ by leveraging PCG and GPUs, they do not address the cubic complexity of generating posterior samples in (3). Moreover, when $n \sim 10^6$, PCG can become prohibitively expensive from both a memory and computational standpoint, as most preconditioners require $\mathcal{O}(nr)$ storage [Cutajar et al., 2016, Avron et al., 2017, Díaz et al., 2023] and each PCG iteration costs $\mathcal{O}(n^2)$.

### A.1.2   SGD-based inference

To address the limitations of PCG-based inference, Lin et al. [2023] proposed to replace PCG with SGD. This reduces iteration cost from $\mathcal{O}(n^2)$ to $\mathcal{O}(bn)$, where $b$ is the gradient batchsize. In addition, they address the challenge of the cubic cost of generating posterior samples by adopting the pathwise conditioning strategy [Wilson et al., 2020, 2021]. Lin et al. [2023] show SGD outperforms PCG and Stochastic Variational Gaussian Processes (SVGP) [Hensman et al., 2013] on various posterior sampling tasks. Lin et al. [2024] further improved upon SGD by introducing Stochastic Dual Descent (SDD). SDD is a coordinate descent algorithm that leverages the dual formulation of the KRR problem to reduce the dependence on the conditioning of $K$ from $\kappa(K(K + \lambda I))$ to $\kappa(K + \lambda I)$. In addition, SDD incorporates momentum and geometric averaging to further enhance performance. Compared to SGD, PCG, and SVGP, Lin et al. [2024] shows that SDD yields the best empirical performance.

While Lin et al. [2023, 2024] address the scaling limitations of PCG and the issue of efficiently sampling from the posterior, there is still room for improvement. In terms of theory, Lin et al. [2023] only establishes that SGD is guaranteed to converge at a reasonable rate along the top eigenvectors of $K$. Convergence along the small eigenvectors is extremely slow, as the rate in Lin et al. [2023] along the $\ell^{th}$ eigenvector depends upon $1/\sqrt{\lambda_i}$. For SDD, the convergence rate is unknown, as Lin et al. [2024] provides no convergence analysis. Aside from unsatisfactory convergence analyses, it is known from optimization theory, that in the worst case, the convergence rate of SGD and SDD is controlled by the condition number [Nemirovski and Yudin, 1983]. As ill-conditioning is a defining property of kernel matrices, this means SDD and SGD will converge slowly. Another price incurred by SDD and SGD for their fast performance is the presence of additional hyperparameters. In particular, the stepsize for these methods must be set appropriately to obtain satisfactory convergence. Tuning the stepsize, while possible, can become expensive for large-scale inference on datasets like taxi. This is in contrast to PCG, which is stepsize-free.

`ADASAP` enjoys the scalability benefits of SDD and SGD, but without their limitations. It only operates on a batch of data at each iteration, so it avoids the $\mathcal{O}(n^2)$ iteration cost of PCG. By incorporating subspace preconditioning, `ADASAP` addresses the ill-conditioning of the kernel matrix to achieve fast

convergence. `ADASAP` also comes with reliable default hyperparameters, which obviates the need for costly hyperparameter tuning.

### A.1.3 Low-dimensional inference

When the dimension $d$ of the data matrix $X$ is low ($d \leq 3$), GP inference can be made far more efficient by exploiting the structure of the kernel matrix and the low-dimensionality of the data. Common approaches in this vein include rank-structured matrices [Ambikasaran et al., 2015, Minden et al., 2017], sparse Cholesky factorization [Chen et al., 2025], and kernel interpolation [Wilson and Nickisch, 2015, Greengard et al., 2025]. In most cases these methods enable $\mathcal{O}(n)$ training time and inference, a massive improvement over standard GP inference methods. This linear time complexity has allowed GP training to scale to datasets with $n$ as large as $10^9$ [Greengard et al., 2025]. Unfortunately, the complexity of these algorithms grows exponentially with the dimension, making them impractical when $d > 3$.

### A.1.4 Variational inference

To address the challenge of large-scale GP inference, many prior works have focused on developing scalable approximate GP methods. The most popular methods are based on variational inference [Titsias, 2009, Hensman et al., 2013, 2015, 2018]. The most well-known approach is SVGP [Hensman et al., 2013], which selects a set of $m$ inducing points by maximizing an evidence lower bound (ELBO). The use of inducing points leads to only need to work with an $n \times m$ kernel matrix instead of the full $n \times n$ kernel matrix. This leads reduces the cost of training to $\mathcal{O}(nm^2 + m^3)$, a significant improvement over exact inference methods when $m$ can be taken to be much smaller than $n$. While SVGP and related algorithms significantly reduce the cost of GPs, they generally exhibit worse performance on inference than methods which make use of the full data [Wang et al., 2019, Lin et al., 2023, 2024]. Thus, it is advisable to use exact inference whenever possible.

## A.2 Sketch-and-project

`ADASAP` builds off the sketch-and-project framework for solving consistent linear systems. Sketch-and-project was first formulated in Gower and Richtárik [2015], who showed that randomized Kaczmarz [Strohmer and Vershynin, 2009], randomized coordinate descent [Leventhal and Lewis, 2010], and randomized Newton [Qu et al., 2016], are all special cases of sketch-and-project. They also established a linear convergence rate for sketch-and-project, which is controlled by the smallest eigenvalue of an expected projection matrix. However, Gower and Richtárik [2015] were unable to provide a fine-grained analysis of sketch-and-project in terms of the spectral properties of the linear system, except for a few special cases.

Follow-up work has combined sketch-and-project with techniques such as Nesterov acceleration [Tu et al., 2017] and extended sketch-and-project to quasi-Newton and Newton type methods Gower et al. [2018, 2019], Hanzely et al. [2020]. Despite these extensions, a sharp convergence analysis of sketch-and-project remained elusive, even in the original setting of consistent linear systems.

Recent work has finally given a sharp analysis of sketch-and-project. Using powerful tools from random matrix theory, Dereziński and Rebrova [2024] provided the first sharp analysis of sketch-and-project when the sketching matrix is sub-Gaussian or sparse. In particular, they were the first to prove that if the matrix exhibits a favorable spectral decay profile, then sketch-and-project exhibits condition number-free convergence. Dereziński and Yang [2024] improve upon these results, showing that by employing a pre-processing step, one can avoid using expensive sketching matrices at each step, and simply sample rows uniformly. They accomplish this by leveraging tools from determinantal point process theory [Mutny et al., 2020, Rodomanov and Kropotov, 2020, Anari et al., 2024]. Dereziński et al. [2024] further refines these results by incorporating Nesterov acceleration and demonstrating an improved computational complexity relative to a fine-grained analysis of PCG—that is, an analysis of PCG that goes beyond the worst-case $\tilde{\mathcal{O}}(\sqrt{\kappa})$ iteration complexity [Trefethen and Bau, 1997].

In addition to improvements in the analysis of exact sketch-and-project for linear systems, significant strides have been made when approximate subspace preconditioning is employed. Dereziński and Yang [2024], Dereziński et al. [2024] both use PCG to approximately solve the linear system defined by the subspace Hessian in each iteration of `SAP`, while Dereziński et al. [2025] augment that system with Tikhonov regularization to further improve its conditioning. Rathore et al. [2025] consider an

approximate sketch-and-project method for solving kernel ridge regression with one right-hand side. They replace $K_{\mathcal{B}\mathcal{B}} + \lambda I$ in SAP with $\hat{K}_{\mathcal{B}\mathcal{B}} + \rho I$, where $\hat{K}_{\mathcal{B}\mathcal{B}}$ is a randomized Nyström approximation of $K_{\mathcal{B}\mathcal{B}}$. They establish linear convergence for the method, with a rate that is comparable to exact SAP when the kernel matrix exhibits an appropriate rate of spectral decay.

Among the above-mentioned works, ADASAP is closest to Rathore et al. [2025], but with significant algorithmic differences such as tail averaging, handling multiple right-hand sides, and distributing bottleneck operations over multiple devices. On the theoretical side, our analysis focuses on the convergence along the top $\ell$-subspace using exact SAP with tail-averaging. In contrast, Rathore et al. [2025] focus on convergence of approximate sketch-and-project (with and without acceleration) *over the entire space*, and as a result, they are only able to show a fast convergence guarantee under certain strong conditions on the spectral decay (effectively limiting the condition number of the problem). Consequently, the global convergence analysis of Rathore et al. [2025] does not explain the rapid initial progress that SAP-style algorithms make on test error. We believe that our two-phase analysis consisting of (i) fast condition number-free sublinear convergence, followed by (ii) a slower global convergence rate dependent upon the subspace condition number, better captures this phenomenon. In particular, our result is the first step in developing a systematic analysis of the generalization properties of SAP-style algorithms.

To the best of our knowledge, our work is the first to systematically investigate the convergence rate of SAP with tail averaging along the top eigenspace. Epperly et al. [2024a] combines tail averaging with randomized Kaczmarz for solving inconsistent linear systems, but they do not investigate the convergence rate along the top spectral subspaces. Steinerberger [2021], Dereziński and Rebrova [2024] look at the convergence rate of the *expected iterates* along a particular eigenvector. However, as noted in Section 3.2, this does not lead to a meaningful convergence rate for the iterates produced by the algorithm, and this also fails to capture the behavior over an entire subspace. Thus, our analysis provides the first concrete characterization of the convergence rate of tail-averaged SAP along the top eigenvectors.

# B Proofs of the Main Results

In this section, we give the detailed proofs for the main theoretical results given in Section 3.2, namely Theorem 3.2 and Corollary 3.3. We start with an informal overview of the analysis in Section B.1, followed by a formal convergence analysis of SAP for solving positive-definite linear systems in Section B.2, and finally adapting the results to GP posterior mean estimation in Section B.3.

Throughout, we adopt the shorthand $A_\lambda := A + \lambda I$, where $A$ is a square matrix, $\lambda \in \mathbb{R}$, and $I$ denotes the identity matrix of the same size as $A$.

## B.1 Overview of the analysis

Let $w_\star := K_\lambda^{-1} y \in \mathbb{R}^n$ be the solution of the linear system defined by $K_\lambda$ and $y$, an let us use $w_t \in \mathbb{R}^n$ to denote the iterates produced by sketch-and-project when solving that system. To convert from the posterior mean error to the Euclidean norm, we first write the spectral basis functions as $u^{(i)} = \frac{1}{\sqrt{\lambda_i}} k(\cdot, X) v_i$, where $(\lambda_i, v_i)$ is the $i^{th}$-largest eigenpair of $K$. Thus, for any $w \in \mathbb{R}^n$ and $m = k(\cdot, X) w \in \mathcal{H}$, we have $\|\text{proj}_\ell(m - m_n)\|_{\mathcal{H}} = \|Q_\ell K^{1/2}(w - w_\star)\| \le \|Q_\ell K_\lambda^{1/2}(w - w_\star)\|$, where $Q_\ell = \sum_{i=1}^{\ell} v_i v_i^T$ is the projection onto the top-$\ell$ subspace of $K$.

Having converted to the Euclidean norm, we express the sketch-and-project update as a recursive formula for its (scaled) residual vector as follows:

$$\Delta_{t+1} := K_\lambda^{1/2}(w_{t+1} - w_\star) = (I - \Pi_{\mathcal{B}}) K_\lambda^{1/2}(w_t - w_\star) = (I - \Pi_{\mathcal{B}}) \Delta_t,$$

where $\Pi_{\mathcal{B}} = K_\lambda^{1/2} I_{\mathcal{B}}^T (K_{\mathcal{B},\mathcal{B}} + \lambda I)^\dagger I_{\mathcal{B}} K_\lambda^{1/2}$ is a random projection defined by $\mathcal{B}$. To characterize the expected convergence of sketch-and-project, we must therefore control the average-case properties of $\Pi_{\mathcal{B}}$. We achieve this by relying on a row sampling technique known as determinantal point processes (DPPs) [Kulesza and Taskar, 2012, Dereziński and Mahoney, 2021]. A $b$-DPP($K_\lambda$) is defined as a distribution over size $b$ index sets $\mathcal{B} \subseteq [n]$ such that $\Pr[\mathcal{B}] \propto \det(K_{\mathcal{B}\mathcal{B}} + \lambda I)$. DPPs can be sampled from efficiently [Calandriello et al., 2020, Anari et al., 2024]: after an initial preprocessing cost of $\tilde{O}(nb^2)$, we can produce each DPP sample of size $b$ in $\tilde{O}(b^3)$ time.

Prior work [Dereziński and Yang, 2024] has shown that when $\mathcal{B}$ is drawn according to $2b$-DPP$(K_\lambda)$, then the expectation of $\Pi_\mathcal{B}$ has the same eigenbasis as $K$. Concretely, we can show that $\bar{\Pi} := \mathbb{E}\,\Pi_\mathcal{B} = V\Lambda V^T$ with $\Lambda = \text{diag}(\bar{\lambda}_1, ..., \bar{\lambda}_n)$, where $\bar{\lambda}_i \geq \frac{1}{1+\phi(b,i)}$ and $V$ consists of the eigenvectors of $K$. This leads to a convergence guarantee for the *expected* residual vector:

$$\|Q_\ell \mathbb{E}\,\Delta_{t+1}\| = \|Q_\ell(I - \bar{\Pi})Q_\ell \mathbb{E}\,\Delta_t\| \leq (1 - \bar{\lambda}_\ell)\|Q_\ell \mathbb{E}\,\Delta_t\|,$$

where we used that $\bar{\Pi}$, $K_\lambda$, and $Q_\ell$ commute with each other. This would suggest that we should obtain a fast linear rate $(1 - \frac{1}{1+\phi(b,\ell)})^t$ for the sketch-and-project iterates in the top-$\ell$ subspace norm. However, it turns out that the actual residual does not attain the same convergence guarantee as its expectation because, unlike $\bar{\Pi}$, the random projection $\Pi_\mathcal{B}$ does not commute with $K_\lambda$ or $Q_\ell$. Indeed,

$$\mathbb{E}\|Q_\ell\Delta_{t+1}\|^2 = \Delta_t^T \mathbb{E}\left[(I - \Pi_\mathcal{B})Q_\ell(I - \Pi_\mathcal{B})\right]\Delta_t,$$

and even though the matrix $(I - \Pi_\mathcal{B})Q_\ell(I - \Pi_\mathcal{B})$ is low-rank, its expectation may not be, since $\Pi_\mathcal{B}$ does not commute with $Q_\ell$. This means that we have no hope of showing a linear convergence rate along the top-$\ell$ subspace. Nevertheless, we are still able to obtain the following bound:

$$\mathbb{E}\left[(I - \Pi_\mathcal{B})Q_\ell(I - \Pi_\mathcal{B})\right] \preceq Q_\ell(I - \bar{\Pi}) + (I - Q_\ell)\bar{\Pi}(I - \bar{\Pi}).$$

This results in a bias-variance decomposition: the first term is the bias, which exhibits fast convergence in top-$\ell$ subspace, while the second term is the variance, which accounts for the noise coming from the orthogonal complement subspace. We then use tail averaging to decay this noise, which allows us to benefit from the fast convergence of expected iterates through a sublinear rate $\frac{\phi(b,\ell)}{t}$.

Finally, to attain the asymptotically faster linear rate for the residual vectors, we observe that every top-$\ell$ subspace norm is upper-bounded by the full norm: $\|Q_\ell\Delta_t\| \leq \|\Delta_t\|$. Thus, we can effectively repeat the above analysis with $\ell = n$ (since $Q_n = I$), in which case the variance term becomes zero, and we recover a linear rate of the form $\left(1 - \frac{1}{1+\phi(b,n)}\right)^t$.

## B.2   Convergence along top-$\ell$ dimensional subspace for positive-definite matrices

In this section, we consider the general problem of solving a linear system $Aw = y$ for an $n \times n$ positive-definite matrix $A$. We provide theoretical results providing the fast convergence guarantees for the iterate sequence generated by SAP along the top-$\ell$ dimensional subspace of $A$, with Theorem B.3 being the main result of this section. We use Theorem B.3 in the next section to derive the subspace convergence result for posterior mean in GP inference, proving Theorem 3.2.

**Preliminaries and notation**   We start by introducing the notation and some results from existing literature for SAP and DPPs. Let $A$ be an $n \times n$ symmetric positive-definite matrix and let $A = VDV^T$ be its eigendecomposition, where $D$ is diagonal with entries $\lambda_1 \geq ... \geq \lambda_n$. For other matrices we use $\lambda_i(M)$ to denote the eigenvalues of $M$. For any vector $v \in \mathbb{R}^n$, define $\|v\|_A := \sqrt{v^T A v}$. Let $V_\ell \in \mathbb{R}^{n \times \ell}$ consist of the top-$\ell$ orthonormal eigenvectors of $A$ and $Q_\ell = V_\ell V_\ell^T$ be the orthogonal projector corresponding to the top-$\ell$ eigenvectors of $A$. We rely on subsampling from fixed-size DPPs [Kulesza and Taskar, 2012, Dereziński and Mahoney, 2021] for our theoretical results. Here, for convenience, we consider the block size to be $2b$, in contrast to $b$ as considered in Algorithm 1. For a fixed $b$ and a positive-definite matrix $A$, we identify $2b$-DPP$(A)$ as a distribution over subsets of $\{1, \ldots, n\}$ of size $2b$, where any subset $\mathcal{B}$ has probability proportional to the corresponding principal submatrix of $A$, i.e., $\det(A_{\mathcal{B},\mathcal{B}})$. The subsampling matrix $S \in \mathbb{R}^{2b \times n}$ corresponding to the set $\mathcal{B}$ is defined as a matrix whose rows are the standard basis vectors associated with the indices in $\mathcal{B}$. For notational convenience, we will sometimes say that $S$ is drawn from $2b$-DPP$(A)$, or $S \sim 2b$-DPP$(A)$, keeping the index set $\mathcal{B}$ implicit.

We use the following result from the literature that provides an exact characterization of the eigenvectors of the expected projection matrix that arises in the analysis of sketch-and-project methods, when using subsampling from fixed-size DPPs.

**Lemma B.1** (Expected projection under $2b$-DPP$(A)$, adapted from Lemma 4.1 [Dereziński and Yang, 2024]). *Let $A \in \mathbb{R}^{n \times n}$ be a symmetric positive-definite matrix with eigenvalues $\lambda_1 \geq \lambda_2 \geq ... \geq \lambda_n$, and let $1 \leq b < n/2$ be fixed. Let $S \sim 2b$-DPP$(A)$. Then, we have*

$$\mathbb{E}[A^{1/2}S^T(SAS^T)^\dagger SA^{1/2}] = VD'V^T,$$

*where $D'$ is a diagonal matrix with $j^{th}$ diagonal entry is lower bounded by $\frac{\lambda_j}{\lambda_j + \frac{1}{b}\sum_{i>b}\lambda_i}$.*

As exact sampling from DPPs is often expensive and requires performing operations as costly as performing the eigendecomposition of $A$ [Kulesza and Taskar, 2012], numerous works have looked at approximate sampling from these distributions [Calandriello et al., 2020, Anari et al., 2024]. Since we exploit the exact characterization of the eigenvectors of the expected projection matrix, our theory requires a very accurate DPP sampling algorithm. Fortunately, the Markov Chain Monte Carlo tools developed by Anari and Dereziński [2020], Anari et al. [2024] allow near-exact sampling from fixed-size DPPs (in the sense that the samples are with high probability indistinguishable from the exact distribution), while offering significant computational gains over exact sampling.

**Lemma B.2** (Sampling from $b$-DPP($A$), adapted from Anari et al. [2024])**.** *Given a positive-definite matrix $A$, there exists an algorithm that draws $t \geq 1$ approximate samples from 2b-DPP(A) in time $O(nb^2 \log^4 n + tb^3 \log^3 n)$. Furthermore, each drawn sample is indistinguishable from an exact sample from 2b-DPP(A) with probability at least $1 - n^{-O(1)}$.*

We now provide the main result of this section, incorporating the above two results in the analysis of sketch-and-project (SAP) along the top-$\ell$ subspace of $A$. Since throughout this section we focus on a linear system with a single right-hand side, we will use $w_t$ instead of $W_t$ to denote the SAP iterates, as these are always vectors. The update rule for SAP is now:

$$w_{t+1} = w_t - S^T (SAS^T)^{-1} S(Aw_t - y)$$

$$\text{or equivalently } A^{1/2}(w_{t+1} - w_\star) = (I - \Pi)A^{1/2}(w_t - w_\star), \tag{5}$$

where $\Pi = A^{1/2} S^T (SAS^T)^\dagger SA^{1/2}$ and $w_\star = A^{-1}y$. Let $\bar{\Pi}$ denote $\mathbb{E}\,\Pi$.

In the rest of the section we prove the following result, which is a slight generalization of Theorem 3.2.

**Theorem B.3** (Fast convergence along top-$\ell$ subspace)**.** *Let $b < n/2$ be fixed and $S \in \mathbb{R}^{2b \times n}$ be a random subsampling matrix sampled from 2b-DPP(A). Furthermore, let $1 \leq \ell < b$ be also fixed and $\lambda_\ell(\bar{\Pi}) \geq 2\lambda_n(\bar{\Pi})$. For any $t > 1$ define $\bar{w}_t = \frac{2}{t} \sum_{i=t/2}^{t-1} w_i$ where $w_i$ are generated using (5). Then,*

$$\mathbb{E}\|A^{1/2}(\bar{w}_t - w_\star)\|_{Q_\ell}^2 \leq \left(1 - \frac{1}{1 + \phi(b, \ell)}\right)^{t/2} \|A^{1/2}(w_0 - w_\star)\|_{Q_\ell}^2$$

$$+ \frac{8}{t} \phi(b, \ell) \left(1 - \frac{1}{2\phi(b, n)}\right)^{t/2} \|w_0 - w_\star\|_A^2,$$

*where $\phi(b, p) = \frac{1}{b} \sum_{i>b} \frac{\lambda_i}{\lambda_p}$.*

The assumption $\lambda_\ell(\bar{\Pi}) \geq 2\lambda_n(\bar{\Pi})$ is not restrictive buts lets us provide a cleaner analysis by avoiding the edge case of $\lambda_\ell(\bar{\Pi}) \approx \lambda_n(\bar{\Pi})$. In fact, in this corner case, we can simply rely on the existing SAP analysis from previous works and recover our main result, Theorem 3.2. For completeness, we derive Theorem B.9 for posterior GP mean inference in Section B.3 without the assumption $\lambda_\ell(\bar{\Pi}) \geq 2\lambda_n(\bar{\Pi})$. The proof of Theorem B.3 appears after the proof of Theorem B.8. We build towards the proof starting with the following result for SAP [Dereziński and Yang, 2024]:

**Lemma B.4** (Linear convergence with SAP)**.** *Let $S \in \mathbb{R}^{2b \times n}$ be a random subsampling matrix sampled from 2b-DPP(A). Then,*

$$\mathbb{E}\|w_{t+1} - w_\star\|_A^2 \leq (1 - \lambda_n(\bar{\Pi}))\mathbb{E}\|w_t - w_\star\|_A^2.$$

However, in the following result, we show that along the top-$\ell$ eigenspace of $A$, the expected iterates can converge at a much faster rate than $1 - \lambda_n(\bar{\Pi})$.

**Lemma B.5** (Convergence of expected iterates along top-$\ell$ subspace)**.** *Let $S \in \mathbb{R}^{2b \times n}$ be a random subsampling matrix sampled from 2b-DPP(A). Then*

$$\|A^{1/2}\mathbb{E}_t[w_{t+1} - w_\star]\|_{Q_\ell}^2 \leq \left(1 - \lambda_\ell(\bar{\Pi})\right)^2 \|A^{1/2}(w_t - w_\star)\|_{Q_\ell}^2,$$

*where $\mathbb{E}_t$ denotes conditional expectation given $w_t$.*

*Proof.* We have

$$Q_\ell A^{1/2}(w_{t+1} - w_\star) = Q_\ell(I - \Pi)A^{1/2}(w_t - w_\star).$$

Taking expectation on both sides and squaring we get,

$$\|A^{1/2}\mathbb{E}_t[w_{t+1} - w_\star]\|_{Q_\ell}^2 = (w_t - w_\star)^T A^{1/2}(I - \bar{\Pi})Q_\ell(I - \bar{\Pi})A^{1/2}(w_t - w_\star).$$

As $A = VDV^T$ we have $\Pi = VD^{1/2}V^TS^T(SVDV^TS^T)^\dagger SVD^{1/2}V^T$. Due to Lemma B.1 we know that $\bar{\Pi}$ and $A$ share the same eigenvectors, therefore, $\bar{\Pi}$ and $Q_\ell$ commute. Consequently,

$$\|A^{1/2}\mathbb{E}_t[w_{t+1} - w_\star]\|_{Q_\ell}^2 = (w_t - w_\star)^T A^{1/2}Q_\ell(I - \bar{\Pi})^2 Q_\ell A^{1/2}(w_t - w_\star)$$
$$\leq (1 - \lambda_\ell(\bar{\Pi}))^2\|A^{1/2}(w_t - w_\star)\|_{Q_\ell}^2.$$

$\square$

We now analyze convergence along the top-$\ell$ dimensional subspace in L2-norm by considering $\mathbb{E}\|A^{1/2}(w_{t+1} - w_\star)\|_{Q_\ell}^2$. We have,

$$\mathbb{E}\|A^{1/2}(w_{t+1} - w_\star)\|_{Q_\ell}^2 = (w_t - w_\star)^T A^{1/2}\mathbb{E}\left[(I - \Pi)Q_\ell(I - \Pi)\right]A^{1/2}(w_t - w_\star).$$

In particular, we need to upper bound $\mathbb{E}\left[(I - \Pi)Q_\ell(I - \Pi)\right]$. We prove the following lemma:

**Lemma B.6** (L2-norm error along top-$\ell$ subspace). *Let $S \in \mathbb{R}^{2b \times n}$ be a random subsampling matrix sampled from 2b-DPP(A). Then*

$$\mathbb{E}\|A^{1/2}(w_{t+1} - w_\star)\|_{Q_\ell}^2 \leq (1 - \lambda_\ell(\bar{\Pi}))\mathbb{E}\|A^{1/2}(w_t - w_\star)\|_{Q_\ell}^2 + \lambda_{\ell+1}(\bar{\Pi}(I - \bar{\Pi}))\mathbb{E}\|A^{1/2}(w_t - w_\star)\|_{I - Q_\ell}^2.$$

*Proof.* First, we rewrite $\mathbb{E}\left[(I - \Pi)Q_\ell(I - \Pi)\right]$ as follows:

$$\mathbb{E}\left[(I - \Pi)Q_\ell(I - \Pi)\right] = Q_\ell - \bar{\Pi}Q_\ell - Q_\ell\bar{\Pi} + \mathbb{E}[\Pi Q_\ell \Pi]$$
$$= (I - \bar{\Pi})Q_\ell - Q_\ell\bar{\Pi} + \mathbb{E}[\Pi Q_\ell \Pi] \tag{6}$$

Crucially, the first two terms terms in (6) live in the top-$\ell$ subspace, but the third term does not. Nevertheless, we are still able to bound it as follows:

$$\mathbb{E}[\Pi Q_\ell \Pi] = \mathbb{E}[\Pi(I - (I - Q_\ell))\Pi] = \mathbb{E}[\Pi] - \mathbb{E}[\Pi(I - Q_\ell)\Pi] \preceq \bar{\Pi} - \bar{\Pi}(I - Q_\ell)\bar{\Pi},$$

where we used Jensen's inequality in the last relation. Substituting in (6) we get,

$$\mathbb{E}\left[(I - \Pi)Q_\ell(I - \Pi)\right] \preceq (I - \bar{\Pi})Q_\ell + (I - Q_\ell)\bar{\Pi}(I - \bar{\Pi}). \tag{7}$$

Using (7) we now upper bound $\mathbb{E}\|A^{1/2}(w_{t+1} - w_\star)\|_{Q_\ell}^2$ as

$$\mathbb{E}\|A^{1/2}(w_{t+1} - w_\star)\|_{Q_\ell}^2 \leq (1 - \lambda_\ell(\bar{\Pi}))\mathbb{E}\|A^{1/2}(w_t - w_\star)\|_{Q_\ell}^2 + \lambda_{\ell+1}(\bar{\Pi}(I - \bar{\Pi}))\mathbb{E}\|A^{1/2}(w_t - w_\star)\|_{I - Q_\ell}^2,$$

which concludes the proof. $\square$

Unrolling this recursive bound, and combining it with the convergence in the full norm, we obtain the following convergence guarantee for SAP iterates without tail averaging.

**Corollary B.7.** *Let $\alpha = \frac{1 - \lambda_\ell(\bar{\Pi})}{1 - \lambda_n(\bar{\Pi})}$. Then,*

$$\mathbb{E}\|A^{1/2}(w_{t+1} - w_\star)\|_{Q_\ell}^2 \leq (1 - \lambda_\ell(\bar{\Pi}))^{t+1}\|A^{1/2}(w_0 - w_\star)\|_{Q_\ell}^2 + \frac{\lambda_{\ell+1}(\bar{\Pi}(I - \bar{\Pi}))}{1 - \alpha}(1 - \lambda_n(\bar{\Pi}))^t\|w_0 - w_\star\|_A^2.$$

*Proof.* Define $\gamma = 1 - \lambda_n(\bar{\Pi})$. Invoking Lemma B.6 we deduce,

$$\mathbb{E}\|A^{1/2}(w_{t+1} - w_\star)\|_{Q_\ell}^2 \leq (1 - \lambda_\ell(\bar{\Pi}))\mathbb{E}\|A^{1/2}(w_t - w_\star)\|_{Q_\ell}^2 + \lambda_{\ell+1}(\bar{\Pi}(I - \bar{\Pi}))\mathbb{E}\|w_t - w_\star\|_A^2.$$

Now, recursively applying the tower rule and Lemma B.4 yields

$$\mathbb{E}\|A^{1/2}(w_{t+1} - w_\star)\|_{Q_\ell}^2 \leq (1 - \lambda_\ell(\bar{\Pi}))\mathbb{E}\|A^{1/2}(w_t - w_\star)\|_{Q_\ell}^2 + \lambda_{\ell+1}(\bar{\Pi}(I - \bar{\Pi}))\gamma^t\|w_0 - w_\star\|_A^2.$$

Unfolding the preceding display yields,

$$\mathbb{E}\|A^{1/2}(w_{t+1} - w_\star)\|_{Q_\ell}^2 \leq (1 - \lambda_\ell(\bar{\Pi}))^{t+1}\|A^{1/2}(w_0 - w_\star)\|_{Q_\ell}^2$$
$$+ \lambda_{\ell+1}(\bar{\Pi}(I - \bar{\Pi}))\left(\sum_{i=0}^t \gamma^i(1 - \lambda_\ell(\bar{\Pi}))^{t-i}\right)\|w_0 - w_\star\|_A^2.$$

Observing that $\sum_{i=0}^{t} \gamma^i(1 - \lambda_\ell(\bar{\Pi}))^{t-i} = \gamma^t \sum_{i=0}^{t} \alpha^{t-i}$ where $\alpha := \frac{1-\lambda_\ell(\bar{\Pi})}{\gamma} < 1$, we have

$$\mathbb{E}\|A^{1/2}(w_{t+1} - w_\star)\|_{Q_\ell}^2 \leq (1 - \lambda_\ell(\bar{\Pi}))^{t+1}\|A^{1/2}(w_0 - w_\star)\|_{Q_\ell}^2 + \frac{\lambda_{\ell+1}(\bar{\Pi}(I - \bar{\Pi}))}{1 - \alpha}\gamma^t\|w_0 - w_\star\|_A^2,$$

which concludes the proof. $\qquad\square$

We now use the tail averaging idea similar to Epperly et al. [2024a], obtaining fast convergence along the top-$\ell$ subspace. The proof of Theorem B.3 is then derived from the following result, after combining it with fixed-size DPP sampling guarantees of Lemma B.1.

**Theorem B.8** (Fast convergence along subspace for tail-averaged iterate). *Let $S \in \mathbb{R}^{2b \times n}$ be a random subsampling matrix sampled from 2b-DPP(A). For any $t > 2$ define $\hat{w}_t = \frac{2}{t}\sum_{i \geq t/2}^{t-1} w_i$ where $w_i$ are generated using update rule* (5). *Then*

$$\mathbb{E}\|A^{1/2}(\hat{w}_t - w_\star)\|_{Q_\ell}^2 \leq (1 - \lambda_\ell(\bar{\Pi}))^{t/2}\|A^{1/2}(w_0 - w_\star)\|_{Q_\ell}^2$$
$$+ 2\frac{\lambda_{\ell+1}(\bar{\Pi}(I - \bar{\Pi}))}{t(1 - \alpha)^2}(1 - \lambda_n(\bar{\Pi}))^{t/2-1}\|w_0 - w_\star\|_A^2,$$

*where* $\alpha = \frac{1-\lambda_\ell(\bar{\Pi})}{1-\lambda_n(\bar{\Pi})}$.

*Proof.* Let $\hat{w}_t = \frac{2}{t}\sum_{i \geq t/2}^{t-1} w_i$. Consider $\mathbb{E}\|A^{1/2}(\hat{w}_t - w_\star)\|_{Q_\ell}^2$, set $t_a = t/2$, and let $\mathbb{E}_{w_r}$ denote the expectation conditioned on $w_r$. Applying the tower rule yields

$$\mathbb{E}\|A^{1/2}(\hat{w}_t - w_\star)\|_{Q_\ell}^2 = \frac{1}{(t - t_a)^2}\sum_{s=t_a}^{t-1}\sum_{r=t_a}^{s} \mathbb{E}[(w_r - w_\star)^T A^{1/2} Q_\ell A^{1/2}(w_s - w_\star)]$$

$$= \frac{1}{(t - t_a)^2}\sum_{s=t_a}^{t-1}\sum_{r=t_a}^{s} \mathbb{E}\left[(w_r - w_\star)^T A^{1/2} Q_\ell \mathbb{E}_{w_r}[Q_\ell A^{1/2}(w_s - w_\star)]\right].$$

For $r < s$, we have

$$\mathbb{E}_{w_r}[Q_\ell A^{1/2}(w_s - w_\star)] = Q_\ell(I - \bar{\Pi})\mathbb{E}_{w_r}[A^{1/2}(w_{s-1} - w_\star)]$$
$$= Q_\ell Q_\ell(I - \bar{\Pi})\mathbb{E}_{w_r}[A^{1/2}(w_{s-1} - w_\star)]$$
$$= Q_\ell(I - \bar{\Pi})Q_\ell\mathbb{E}_{w_r}[A^{1/2}(w_{s-1} - w_\star)].$$

Recursing on the above relation yields

$$\mathbb{E}_{w_r}[Q_\ell A^{1/2}(w_m - w_\star)] = Q_\ell(I - \bar{\Pi})^{m-r} Q_\ell[Q_\ell A^{1/2}(w_r - w_\star)].$$

Therefore,

$$\mathbb{E}\|A^{1/2}(\hat{w}_t - w_\star)\|_{Q_\ell}^2 = \frac{1}{(t - t_a)^2}\sum_{s=t_a}^{t-1}\sum_{r=t_a}^{s} \mathbb{E}\left[(w_r - w_\star)^T A^{1/2} Q_\ell(I - \bar{\Pi})^{s-r} Q_\ell A^{1/2}(w_r - w_\star)\right]$$

$$\leq \frac{1}{(t - t_a)^2}\sum_{s=t_a}^{t-1}\sum_{r=t_a}^{s} (1 - \lambda_\ell(\bar{\Pi}))^{s-r}\mathbb{E}\|A^{1/2}(w_r - w_\star)\|_{Q_\ell}^2.$$

Applying Lemma B.6 to upper bound $\mathbb{E}\|A^{1/2}(w_r - w_\star)\|_{Q_\ell}^2$ obtains

$$\mathbb{E}\|A^{1/2}(\hat{w}_t - w_\star)\|_{Q_\ell}^2 \leq \frac{1}{(t - t_a)^2}\|A^{1/2}(w_0 - w_\star)\|_{Q_\ell}^2 \sum_{s=t_a}^{t-1}\sum_{r=t_a}^{s}(1 - \lambda_\ell(\bar{\Pi}))^s$$

$$+ \frac{1}{(t - t_a)^2}\frac{\lambda_{\ell+1}(\bar{\Pi}(I - \bar{\Pi}))}{1 - \alpha}\|A^{1/2}(w_0 - w_\star)\|^2 \sum_{s=t_a}^{t-1}\sum_{r=t_a}^{s}(1 - \lambda_\ell(\bar{\Pi}))^{s-r}\gamma^{r-1}$$

$$\leq (1 - \lambda_\ell(\bar{\Pi}))^{t_a}\|A^{1/2}(w_0 - w_\star)\|_{Q_\ell}^2$$

$$+ \gamma^{t_a-1}\frac{\lambda_{\ell+1}(\bar{\Pi}(I - \bar{\Pi}))}{1 - \alpha}\|A^{1/2}(w_0 - w_\star)\|^2 \frac{1}{(t - t_a)^2}\sum_{s=t_a}^{t-1}\sum_{r=t_a}^{s}\alpha^{s-r},$$

where $\gamma = 1 - \lambda_n(\bar{\Pi})$.

Using $\sum_{s=t_a}^{t-1} \sum_{r=t_a}^{s} \alpha^{s-r} < (t - t_a) \sum_{i=0}^{\infty} \alpha^i = \frac{t-t_a}{1-\alpha}$ we get,

$$\mathbb{E}\|A^{1/2}(\hat{w}_t - w_\star)\|_{Q_\ell}^2 \leq (1 - \lambda_\ell(\bar{\Pi}))^{t_a} \|A^{1/2}(w_0 - w_\star)\|_{Q_\ell}^2 + \frac{\lambda_{\ell+1}(\bar{\Pi}(I - \bar{\Pi}))}{(t - t_a)(1 - \alpha)^2} \gamma^{t_a - 1} \|A^{1/2}(w_0 - w_\star)\|^2.$$

Substituting $t_a = t/2$, the result immediately follows. $\qquad\square$

**Completing the proof of Theorem B.3**  It remains to use the guarantees for the eigenvalues of the matrix $\bar{\Pi}$ from Lemma B.1.

*Proof of Theorem B.3.*  Recalling the definition of $\alpha$ from Theorem B.8 and our assumption that $\lambda_\ell(\bar{\Pi}) \geq 2\lambda_n(\bar{\Pi})$, we obtain for $\gamma = 1 - \lambda_n(\bar{\Pi})$ that $(1 - \alpha)^{-1} < 2\gamma/\lambda_\ell(\bar{\Pi})$. Consequently,

$$\begin{aligned}
\frac{\lambda_{\ell+1}(\bar{\Pi}(I - \bar{\Pi}))}{(1 - \alpha)^2} &\leq \frac{\lambda_\ell(\bar{\Pi}(I - \bar{\Pi}))}{(1 - \alpha)^2} \\
&< \frac{4(1 - \lambda_\ell(\bar{\Pi}))}{\lambda_\ell(\bar{\Pi})} \gamma^2 \\
&< 4 \left( \frac{1}{\lambda_\ell(\bar{\Pi})} - 1 \right) \gamma \\
&\overset{(1)}{<} 4 \left( \frac{1}{b} \sum_{i>b} \frac{\lambda_i}{\lambda_\ell} \right) \gamma \\
&\overset{(2)}{=} 4\phi(b, \ell)\gamma,
\end{aligned}$$

where (1) applies Lemma B.1 and (2) defines $\phi(b, \ell) = \frac{1}{b} \sum_{i>b} \frac{\lambda_i}{\lambda_\ell}$. Observing the elementary inequalities:

$$\gamma < 1 - \frac{1}{2\phi(b, n)}, \quad 1 - \lambda_\ell(\bar{\Pi}) \leq \left( 1 - \frac{1}{1 + \phi(b, \ell)} \right),$$

we immediately deduce from Theorem B.8 that

$$\begin{aligned}
\mathbb{E}\|A^{1/2}(\hat{w}_t - w_\star)\|_{Q_\ell}^2 &\leq \left( 1 - \frac{1}{1 + \phi(b, \ell)} \right)^{t/2} \|A^{1/2}(w_0 - w_\star)\|_{Q_\ell}^2 \\
&\quad + \frac{8}{t} \phi(b, \ell) \left( 1 - \frac{1}{2\phi(b, n)} \right)^{t/2} \|w_0 - w_\star\|_A^2.
\end{aligned}$$

$\qquad\square$

## B.3  Posterior mean inference along top-$\ell$ subspace for GPs

We begin by providing some background on GP inference in the Hilbert space setting. This allows for graceful transition from Hilbert space norm over the posterior mean to vector norms over $\mathbb{R}^n$. We recall that $f$ is a Gaussian process and $\{(x_i, y_i)\}_{i=1}^n$ represents the training data. The posterior Gaussian process is characterized by $\mathcal{N}(m_n(\cdot), k_n(\cdot, \cdot))$, where

$$\begin{aligned}
m_n(\cdot) &= m(\cdot) + k(\cdot, X)(K + \lambda I)^{-1} y, \\
k_n(\cdot, \cdot) &= k(\cdot, \cdot) - k(\cdot, X)(K + \lambda I)^{-1} k(X, \cdot).
\end{aligned}$$

Let $\mathcal{H}$ be the reproducing kernel Hilbert space (RKHS) associated with the kernel $k(\cdot, \cdot)$. Assuming $m(\cdot) = 0$, the mean function $m_n(\cdot)$ can be identified as an element of the subspace $\mathcal{H}_n$ defined as

$$\mathcal{H}_n := \left\{ \sum_{i=1}^n w_i k(\cdot, x_i) \mid w \in \mathbb{R}^n \right\}.$$

In particular, $m_n = \sum_{i=1}^n (w_\star)_i k(\cdot, x_i)$, where $w_\star = (K + \lambda I)^{-1} y$. Furthermore, note that for any element $m' \in \mathcal{H}_n$, we have $\|m'\|_{\mathcal{H}}^2 = w^T K w = \|w\|_K^2$. The operator $\mathcal{C}_n :=$

$\frac{1}{n}\sum_{i=1}^{n}k(\cdot,x_i)\otimes k(\cdot,x_i)$ is known as the empirical covariance operator. Let $v_j$ denote the $j^{th}$ unit eigenvector of $K$ with eigenvalue $\lambda_j$. It is straightforward to show that $u_j := \frac{1}{\sqrt{\lambda_j}}\sum_{i=1}^{n}v_{ji}k(\cdot,x_i)$ is a unit eigenvector of the unnormalized empirical covariance operator $\sum_{i=1}^{n}k(\cdot,x_i)\otimes k(\cdot,x_i)$ with eigenvalue $\lambda_j$. Let $V_\ell \in \mathbb{R}^{n\times\ell}$ consists of top-$\ell$ orthogonal eigenvectors of $K$ as columns and $Q_\ell = V_\ell V_\ell^T$ be a projection matrix onto the subspace spanned by $v_1,\dots,v_\ell$. Consider the $\ell$-dimensional subspace $\mathcal{H}_\ell$ defined as

$$\mathcal{H}_\ell := \left\{\sum_{i=1}^{n}(Q_\ell w)_i k(\cdot,x_i) \mid w \in \mathbb{R}^n\right\}.$$

We claim that $\mathcal{H}_\ell$ is the subspace formed by top-$\ell$ eigenvectors of the empirical covariance operator. This can be seen clearly by choosing $w = v_j$ for $1 \leq j \leq \ell$, we get $u_j \in \mathcal{H}_\ell$. We have the following conclusions.

- For any $m' = \sum_{i=1}^{n}w_i k(\cdot,x_i)$, we have $\|m' - m_n\|_{\mathcal{H}}^2 = \|w - w_\star\|_K^2$.

- The element $m'_{Q_\ell} := \sum_{i=1}^{n}(Q_\ell w)_i k(\cdot,x_i)$ is an orthogonal projection of $m'$ onto $\mathcal{H}_\ell$. This can be seen as

$$m' = \sum_{i=1}^{n}w_i k(\cdot,x_i) = \underbrace{\sum_{i=1}^{n}(Q_\ell w)_i\, k(\cdot,x_i)}_{m'_{Q_\ell}} + \underbrace{\sum_{i=1}^{n}((I-Q_\ell)w)_i\, k(\cdot,x_i)}_{m'_{Q_\ell^c}}.$$

  and finally noting that $\langle m'_{Q_\ell}, m'_{Q_\ell^c}\rangle_{\mathcal{H}} = w^T(I - Q_\ell)KQ_\ell w = 0$.

- For any $m' = \sum_{i=1}^{n}w_i k(\cdot,x_i)$, we have $\|\text{proj}_\ell(m') - \text{proj}_\ell(m_n)\|_{\mathcal{H}}^2 = \|Q_\ell(w - w_\star)\|_K^2$, where $\text{proj}_\ell(m')$ denotes orthogonal projection of $m'$ onto $\mathcal{H}_\ell$.

We now derive the main result of this section by using Theorem B.3. We replace $A$ by $K_\lambda$ in the statement of Theorem B.3 and use $\lambda_i'$ to denote the $i^{th}$ eigenvalue of $K_\lambda$, i.e., $\lambda_i + \lambda$ where $\lambda_i$ is the $i$th eigenvalue of $K$. Furthermore, Theorem 3.2 can be derived from the following result by noticing that sampling from 2b-DPP($K_\lambda$) costs time $O(nb^2\log^4 n)$ for preprocessing and an additional $O(b^3\log^3 n)$ for actual sampling at every iteration (see Lemma B.2). Here is the main result of the section:

**Theorem B.9** (Subspace convergence for GP inference). *Let $b < n/2$ and $S \in \mathbb{R}^{2b\times n}$ be a random subsampling matrix sampled from 2b-DPP($K_\lambda$). For any $t > 2$ define $\hat{w}_t = \frac{2}{t}\sum_{i=t/2}^{t-1}w_i$ where $w_i$ are generated using the update rule (5). Then, sketch-and-project initialized at 0 satisfies*

$$\mathbb{E}\|\text{proj}_\ell(\hat{m}_t) - \text{proj}_\ell(m_n)\|_{\mathcal{H}}^2 \leq \min\left\{\frac{8\phi(b,\ell)}{t}, \left(1 - \frac{1}{2\phi(b,n)}\right)^{t/2}\right\}\|y\|_{K_\lambda^{-1}}^2.$$

*where $\hat{m}_t = \sum_{i=1}^{n}\hat{w}_{ti}k(\cdot,w_i)$ and $\phi(b,p) = \frac{1}{b}\sum_{i>b}\frac{\lambda_i+\lambda}{\lambda_p+\lambda}$.*

*Proof.* If $\lambda_\ell(\bar{\Pi}) \geq 2\lambda_n(\bar{\Pi})$, then we rely on Theorem B.3 and replace $A$ by $K_\lambda$. We have the following observation: For any $t > 2$, we have

$$\left(1 - \frac{1}{1+\phi(b,\ell)}\right)^{t/2} = \left(\frac{\phi(b,\ell)}{1+\phi(b,\ell)}\right)^{t/2} \leq \frac{2\phi(b,\ell)}{t}.$$

Therefore for all $t > 2$, we have,

$$\mathbb{E}\|K_\lambda^{1/2}(\hat{w}_t - w_\star)\|_{Q_\ell}^2 \leq \min\left\{\frac{8\phi(b,\ell)}{t}, \left(1 - \frac{1}{2\phi(b,n)}\right)^{t/2}\right\}\|K_\lambda^{1/2}w_\star\|^2.$$

Let $\hat{m}_t = \sum_{i=1}^{n}\hat{w}_{ti}k(\cdot,x_i)$ where $\hat{w}_{ti}$ denote $i^{th}$ coordinate of $\hat{w}_t$ and $m_n = \sum_{i=1}^{n}w_{\star i}k(\cdot,x_i)$ where $w_\star = K_\lambda^{-1}y$. Then we have,

$$\mathbb{E}\|\text{proj}_\ell(\hat{m}_t) - \text{proj}_\ell(m_n)\|_{\mathcal{H}}^2 = \mathbb{E}\|Q_\ell(\hat{w}_t - w_\star)\|_K^2$$
$$\leq \mathbb{E}\|Q_\ell(\hat{w}_t - w_\star)\|_{K_\lambda}^2$$
$$\leq \min\left\{\frac{8\phi(b,\ell)}{t}, \left(1 - \frac{1}{2\phi(b,n)}\right)^{t/2}\right\}\|K_\lambda^{1/2}w_\star\|^2$$
$$= \min\left\{\frac{8\phi(b,\ell)}{t}, \left(1 - \frac{1}{2\phi(b,n)}\right)^{t/2}\right\}\|y\|_{K_\lambda^{-1}}^2.$$

On the other hand if $\lambda_\ell(\bar{\Pi}) < 2\lambda_n(\bar{\Pi})$, then we simply use the SAP analysis and get

$$\mathbb{E}\|Q_\ell(\hat{w}_t - w_*)\|_{K_\lambda}^2 \leq \mathbb{E}\|\hat{w}_t - w_*\|_{K_\lambda}^2 = \frac{4}{t^2} \cdot \mathbb{E}\left\|\sum_{i=t/2}^{t-1}(w_i - w_*)\right\|_{K_\lambda}^{1/2}$$

$$= \frac{4}{t^2}\left(\sum_{i=t/2}^{t-1}\mathbb{E}\|w_i - w_*\|_{K_\lambda}^2 + \sum_{s,r=t/2,s>r}^{t-1}\mathbb{E}(w_s - w_*)K_\lambda(w_r - w_*)\right)$$

$$\leq \left(1 - \frac{1}{2\phi(b,n)}\right)^{t/2}\|K_\lambda^{1/2}w_*\|^2,$$

where the last inequality can be obtained using a recursive argument similar to Theorem B.8 by plugging in linear convergence guarantees for SAP (using Lemma B.4 and Lemma B.5 with $\ell = n$). We get,

$$\mathbb{E}\|\text{proj}_\ell(\hat{m}_t) - \text{proj}_\ell(m_n)\|_{\mathcal{H}}^2 \leq \mathbb{E}\|Q_\ell(\hat{w}_t - w_\star)\|_{K_\lambda}^2 \leq \left(1 - \frac{1}{2\phi(b,n)}\right)^{t/2}\|K_\lambda^{1/2}w_*\|^2$$

$$= \min\left\{\frac{4\phi(b,n)}{t}, \left(1 - \frac{1}{2\phi(b,n)}\right)^{t/2}\right\}\|y\|_{K_\lambda^{-1}}^2$$

$$< \min\left\{\frac{8\phi(b,\ell)}{t}, \left(1 - \frac{1}{2\phi(b,n)}\right)^{t/2}\right\}\|y\|_{K_\lambda^{-1}}^2.$$

Combining the results for both scenarios: $\lambda_\ell(\bar{\Pi}) \geq 2\lambda_n(\bar{\Pi})$ or $\lambda_\ell(\bar{\Pi}) < 2\lambda_n(\bar{\Pi})$, we conclude the proof. $\square$

**Time complexity analysis** We now use the above guarantee to provide the time complexity analysis for estimating the GP posterior mean. The following result immediately implies Corollary 3.3.

**Corollary B.10.** *Suppose that the matrix $K$ exhibits polynomial spectral decay, i.e., $\lambda_i(K) = \Theta(i^{-\beta})$ for some $\beta > 1$. Then for any $\ell \in \{1, \ldots, n\}$, $\lambda = O(1)$ and $\epsilon \in (0,1)$, choosing $b = 2\ell$ we can find $\hat{m}$ that with probability at least $0.99$ satisfies $\|\text{proj}_\ell(\hat{m}) - \text{proj}_\ell(m_n)\|_{\mathcal{H}}^2 \leq \epsilon\|\text{proj}_\ell(m_n)\|_{\mathcal{H}}^2$ in*

$$\mathcal{O}\left(n\ell^2\log^4 n + (n^2 + n\ell^2\log^3 n)\min\left\{\frac{\log(n/\ell)}{\epsilon}, \left(1 + \frac{\ell(\lambda_\ell(K)+\lambda)}{n(\lambda_n(K)+\lambda)}\right)\log(n/\ell\epsilon)\right\}\right) \quad time.$$

*Proof.* As $y = f(X) + g$, where $g \sim \mathcal{N}(0, \lambda I)$ and $f(X) \sim \mathcal{N}(0, K)$, we have $y \sim \mathcal{N}(0, K_\lambda)$. This implies $\mathbb{E}[yy^T] = K_\lambda$. Therefore, $\mathbb{E}\|K_\lambda^{1/2}w_\star\|^2 = n$ and $\mathbb{E}\|Q_\ell K_\lambda^{1/2}w_\star\|^2 = \ell$. So, using standard Gaussian concentration of measure, it follows that with probability $0.999$, $\|K_\lambda^{1/2}w_\star\|^2 \leq \mathcal{O}(1)\frac{n}{\ell}\|K_\lambda^{1/2}w_\star\|_{Q_\ell}^2$. Furthermore,

$$\|K_\lambda^{1/2}w_\star\|_{Q_\ell}^2 \leq \left(1 + \frac{\lambda}{\lambda_\ell}\right)\|K^{1/2}w_\star\|_{Q_\ell}^2 = \left(1 + \frac{\lambda}{\lambda_\ell}\right)\|\text{proj}_\ell(m_n)\|_{\mathcal{H}}^2.$$

We get the following relation:

$$\mathbb{E}\|\mathrm{proj}_\ell(\hat{m}_t) - \mathrm{proj}_\ell(m_n)\|_{\mathcal{H}}^2 \le \mathcal{O}(1)\left[\frac{n(1+\lambda/\lambda_\ell)}{\ell}\min\left\{\frac{\phi(b,\ell)}{t}, \left(1 - \frac{1}{4\phi(b,n)}\right)^{t/2}\right\}\right]\|\mathrm{proj}_\ell(m_n)\|_{\mathcal{H}}^2$$

Now let $b > \ell + \sum_{i>b}\lambda_i/(\lambda_\ell + \lambda)$. We have $\phi(b,\ell) \le \frac{1}{b}\sum_{i>b}\frac{\lambda_i}{\lambda_\ell + \lambda} + \frac{n\lambda}{b(\lambda_\ell + \lambda)} \le 1 + \frac{n\lambda}{b(\lambda_\ell + \lambda)}$.
We consider following two cases:

*Case 1:* $\frac{n\lambda}{b(\lambda_\ell + \lambda)} \le 1$. In this case we get $\phi(b,\ell) < 2$. After $t = O(\frac{n}{\ell\epsilon})$ iterations we get $\mathbb{E}\|\mathrm{proj}_\ell(\hat{m}_t) - \mathrm{proj}_\ell(m_n)\|_{\mathcal{H}}^2 \le \epsilon\|\mathrm{proj}_\ell(m_n)\|_{\mathcal{H}}^2$.

*Case 2:* $\frac{n\lambda}{b(\lambda_\ell + \lambda)} > 1$. We get $\lambda > \frac{b\lambda_\ell}{n}$. In this case we have $\phi(b,n) = \frac{1}{b}\sum_{i>b}\frac{\lambda_i + \lambda}{\lambda_n + \lambda} < \frac{n}{b} + \frac{1}{b}\sum_{i>b}\frac{\lambda_i}{\lambda} < \frac{2n}{b}$. Therefore, after $t = \mathcal{O}\left(\frac{n\log\left(\frac{n(1+\lambda/\lambda_\ell)}{\ell\epsilon}\right)}{b}\right)$ iterations, we get $\mathbb{E}\|\mathrm{proj}_\ell(\hat{m}_t) - \mathrm{proj}_\ell(m_n)\|_{\mathcal{H}}^2 \le \epsilon\|\mathrm{proj}_\ell(m_n)\|_{\mathcal{H}}^2$.

On the other hand, in either case

$$\phi(b,n) = \frac{1}{b}\sum_{i>b}\frac{\lambda_i + \lambda}{\lambda_n + \lambda} \le \frac{n}{b}\left(1 + \frac{(\lambda_\ell + \lambda)}{(\lambda_n + \lambda)}\frac{1}{n}\sum_{i>b}\frac{\lambda_i}{\lambda_\ell + \lambda}\right) < \frac{n}{b}\left(1 + \frac{b}{n}\frac{\lambda_\ell + \lambda}{\lambda_n + \lambda}\right),$$

as we assumed $b > \sum_{i>b}\frac{\lambda_i}{\lambda_\ell + \lambda}$. Furthermore, for given spectral decay for any $\beta > 1$, we have $\ell + \sum_{i>b}\frac{\lambda_i}{\lambda_\ell} < 2\ell$. This implies for $b = 2\ell$ we obtain $\epsilon$ accuracy result in $t = \mathcal{O}\left(\frac{n}{\ell}\min\left\{\frac{\log(n/\ell)}{\epsilon}, \left(1 + \frac{\ell}{n}\frac{\lambda_\ell + \lambda}{\lambda_n + \lambda}\right)\log(n/\ell\epsilon)\right\}\right)$, where we used $\lambda = \mathcal{O}(1)$ to get $\log\left(\frac{n}{\ell}(1 + \lambda/\lambda_\ell)\right) = \mathcal{O}\left(\log(\frac{n}{\ell})\right)$. Then, after applying Markov's inequality, $\hat{m} = \hat{m}_t$ obtains the desired $\epsilon$ accuracy with probability 0.999. The total time complexity follows by combining the number of iterations with cost of approximate sampling from $2b$-DPP$(A)$ (Lemma B.2) and additional per iteration cost of $\mathcal{O}(n\ell + \ell^3\log^3 n)$. Note that while the sampling algorithm is not exact, taking the union bound with respect to the high probability guarantees in Lemma B.2 ensures that all of the samples are indistinguishable from true DPP with probability 0.999. Finally, union bounding over the three 0.999 probability events we have invoked concludes the proof. $\qquad\square$

**Improved guarantee for very ill-conditioned problems** Here, we show that when $\lambda$ is very small (i.e., $K_\lambda$ is highly ill-conditioned), then we can obtain an even better time complexity in the first phase of the convergence, addressing the claim in Remark 3.4.

**Corollary B.11.** *Suppose that the matrix $K$ exhibits polynomial spectral decay, i.e., $\lambda_i(K) = \Theta(i^{-\beta})$ for some $\beta > 1$ and $\lambda < \frac{1}{nb^{\beta-1}}$, then we can find $\hat{m}$ that with probability at least 0.99 satisfies $\|\mathrm{proj}_\ell(\hat{m}) - \mathrm{proj}_\ell(m_n)\|_{\mathcal{H}}^2 \le \epsilon\|\mathrm{proj}_\ell(m_n)\|_{\mathcal{H}}^2$ in*

$$\mathcal{O}\left(nb^2\log^4 n + (n^2 + nb^2\log^3 n)(\ell/b)^{\beta-1}/\epsilon\right).$$

*Proof.* We reconsider case 1 from the proof of Corollary B.10. Using the given spectral decay profile for $K$ and that $\lambda < \frac{1}{nb^{\beta-1}}$, we get $\phi(b,\ell) = \mathcal{O}\left(\frac{b^{-\beta}}{\ell^{-\beta}}\right)$. Therefore after $t = \frac{n}{\ell\epsilon}\frac{b^{-\beta}}{\ell^{-\beta}}$ we obtain the $\epsilon$ approximation guarantee. The total runtime complexity becomes:

$$\mathcal{O}\left(nb^2\log^4 n + (n^2 + nb^2\log^3 n)(\ell/b)^{\beta-1}/\epsilon\right).$$

$\qquad\square$

# C  Additional Algorithmic Details for `ADASAP`

We provide additional implementation details for `ADASAP`. Appendix C.1 describes how we distribute matrix-matrix products across rows and columns in the algorithms `ColDistMatMat` and `RowDistMatMat`. Appendix C.2 describes the practical implementation of the randomized Nyström approximation and provides pseudocode for `RandNysAppx`. Appendix C.3 describes how we compute

preconditioned smoothness constants and provides pseudocode for `GetStepsize`. Appendix C.4 provides pseudocode for Nesterov acceleration in `NestAcc`. Appendix C.5 provides a scaling plot illustrating the speedups achieved by using multiple GPUs in `ADASAP`. Appendix C.6 investigates the impact of tail averaging on the performance of `ADASAP`.

All operations involving kernel matrices are performed using pykeops [Charlier et al., 2021], which allows us to avoid instantiating kernel matrices explicitly in memory. To see the full details of our implementation, we recommend the reader to view our codebase.

## C.1 Distributed matrix-matrix products

Here, we provide details for how we implement the distributed matrix-matrix products in `ADASAP`. `ColDistMatMat` (Algorithm 3) shows how we distribute the matrix-matrix product $K_{\mathcal{B}n}W_t$ in `ADASAP` and `RowDistMatMat` (Algorithm 4) shows how we distribute the calculation of the Nyström sketch in `ADASAP`. Our implementation of these algorithms uses torch.multiprocessing to spawn a CUDA context on each device (i.e., a worker) and uses pykeops to generate the column and row block oracles.

---

**Algorithm 3** `ColDistMatMat`

---

**Require:** Right-hand side matrix $W \in \mathbb{R}^{n \times n_{\text{rhs}}}$, row indices $\mathcal{B}$, workers $\{\mathcal{W}_1, \dots, \mathcal{W}_{\text{N}_{\text{workers}}}\}$
    Partition rows of $W$ as $\{W_1, \dots, W_{\text{N}_{\text{workers}}}\}$
    Send $W_i$ to $\mathcal{W}_i$
    Generate column block oracle $\mathcal{K}_{\mathcal{W}_i}^{\text{col}}$ using $\mathcal{B}$
    $(K_{\mathcal{B}n}W)_i \leftarrow \mathcal{K}_{\mathcal{W}_i}^{\text{col}}[W_i]$                ▷ Compute column block products in parallel
    Aggregate $K_{\mathcal{B}n}W \leftarrow \sum_{i=1}^{\text{N}_{\text{workers}}} (K_{\mathcal{B}n}W)_i$
    **return** $K_{\mathcal{B}n}W$

---

**Algorithm 4** `RowDistMatMat`

---

**Require:** Right-hand side matrix $\Omega \in \mathbb{R}^{n \times r}$, row indices $\mathcal{B}$, workers $\{\mathcal{W}_1, \dots, \mathcal{W}_{\text{N}_{\text{workers}}}\}$
    Send $\Omega$ to each $\mathcal{W}_i$
    Generate row block oracle $\mathcal{K}_{\mathcal{W}_i}^{\text{row}}$ using $\mathcal{B}$
    $(K_{\mathcal{B}\mathcal{B}}\Omega)_i \leftarrow \mathcal{K}_{\mathcal{W}_i}^{\text{row}}[\Omega]$              ▷ Compute row block products in parallel
    Aggregate $K_{\mathcal{B}\mathcal{B}}\Omega \leftarrow \left[ (K_{\mathcal{B}\mathcal{B}}\Omega)_1^T \ \dots \ (K_{\mathcal{B}\mathcal{B}}\Omega)_{\text{N}_{\text{workers}}}^T \right]^T$
    **return** $K_{\mathcal{B}\mathcal{B}}\Omega$

---

## C.2 Randomized Nyström approximation

Here, we present a practical implementation of the Nyström approximation used in `ADASAP` (Algorithm 2) in `RandNysAppx` (Algorithm 5). The Randomized Nyström approximation of $K_{\mathcal{B}\mathcal{B}}$ with test matrix $\Omega \in \mathbb{R}^{b \times r}$ [Tropp et al., 2017] is given by:

$$\hat{K}_{\mathcal{B}\mathcal{B}} = (K_{\mathcal{B}\mathcal{B}}\Omega)(\Omega^T K_{\mathcal{B}\mathcal{B}}\Omega)^{\dagger}(K_{\mathcal{B}\mathcal{B}}\Omega)^T.$$

The preceding formula is numerically unstable, so `ADASAP` uses `RandNysAppx`, which is based on Tropp et al. [2017, Algorithm 3]. eps($x$) is defined as the positive distance between $x$ and the next largest floating point number of the same precision as $x$. The resulting Nyström approximation $\hat{M}$ is given by $U S U^T$, where $U \in \mathbb{R}^{p \times r}$ is an orthogonal matrix that contains the approximate top-$r$ eigenvectors of $M$, and $S \in \mathbb{R}^r$ contains the top-$r$ eigenvalues of $M$. The Nyström approximation is positive-semidefinite but may have eigenvalues that are equal to $0$. In our algorithms, this approximation is always used in conjunction with a regularizer to ensure positive definiteness.

The dominant cost in Algorithm 5 is computing the SVD of $B$ at a cost of $\mathcal{O}(pr^2)$. This is the source of the $\mathcal{O}(pr^2)$ cost in Phase III of `ADASAP`.

---

**Algorithm 5** `RandNysAppx`

---

**Require:** sketch $M\Omega \in \mathbb{R}^{p \times r}$, sketching matrix $\Omega \in \mathbb{R}^{p \times r}$, approximation rank $r \le p$

$\quad \Delta \leftarrow \text{eps}(\Omega^T M\Omega.\text{dtype}) \cdot \text{Tr}(\Omega^T M\Omega)$          ▷ Compute shift for stability

$\quad C \leftarrow \text{chol}(\Omega^T M\Omega + \Delta\Omega^T\Omega)$     ▷ Cholesky decomposition: $C^T C = \Omega^T M\Omega + \Delta\Omega^T\Omega$

$\quad B \leftarrow YC^{-1}$                                           ▷ Triangular solve

$\quad [U, \Sigma, \sim] \leftarrow \text{svd}(B, 0)$                                   ▷ Thin SVD

$\quad S \leftarrow \max\{0, \text{diag}(S^2 - \Delta I)\}$     ▷ Compute eigs, and remove shift with element-wise max

$\quad$ **return** $U, S$

---

### C.2.1 Applying the Nyström approximation to a vector

In our algorithms, we often perform computations of the form $(\hat{M} + \rho I)^{-1}g = (USU^T + \rho I)^{-1}g$. This computation can be performed in $\mathcal{O}(rp)$ time using the Woodbury formula [Higham, 2002]:

$$(USU^T + \rho I)^{-1}g = U(S + \rho I)^{-1} U^T g + \frac{1}{\rho}(g - UU^T g). \tag{8}$$

We also use the randomized Nyström approximation to compute preconditioned smoothness constants in `GetStepsize` (Algorithm 6). This computation requires the calculation $(P + \rho I)^{-1/2}v$ for some $v \in \mathbb{R}^p$, which can also be performed in $\mathcal{O}(pr)$ time using the Woodbury formula:

$$(USU^T + \rho I)^{-1/2}v = U(S + \rho I)^{-1/2} U^T v + \frac{1}{\sqrt{\rho}}(v - UU^T v). \tag{9}$$

In single precision, (8) is unreliable for computing $(P + \rho I)^{-1}g$. This instability arises due to roundoff error: the derivation of (8) assumes that $\hat{U}^T \hat{U} = I$, but we have empirically observed that orthogonality does not hold in single precision. To improve stability, we compute a Cholesky decomposition $LL^T$ of $\rho S^{-1} + U^T U$, which takes $\mathcal{O}(pr^2)$ time since we form $U^T U$. Using the Woodbury formula and Cholesky factors,

$$
\begin{aligned}
(USU^T + \rho I)^{-1}g &= \frac{1}{\rho}g - \frac{1}{\rho}U(\rho S^{-1} + U^T U)^{-1}U^T g \\
&= \frac{1}{\rho}g - \frac{1}{\rho}UL^{-T}L^{-1}U^T g.
\end{aligned}
$$

This computation can be performed in $\mathcal{O}(pr)$ time, since the $\mathcal{O}(r^2)$ cost of triangular solves with $L^T$ and $L$ is negligible compared to the $\mathcal{O}(pr)$ cost of multiplication with $U^T$ and $U$.

Unlike Eq. (8), we find using (9) in `GetStepsize` yields excellent performance, i.e., we do not need to perform any additional stabilization.

### C.3 Computing the stepsize

Here, we provide the details of the `GetStepsize` procedure in `ADASAP` (Algorithm 2) for automatically computing the stepsize. Our procedure is inspired by Rathore et al. [2025], who show that approximate `SAP` with the Nyström approximation converges when

$$\eta_{\mathcal{B}} = 1/\lambda_1 \left( (P_{\mathcal{B}} + \rho I)^{-1/2}(K_{\mathcal{B}\mathcal{B}} + \lambda I)(P_{\mathcal{B}} + \rho I)^{-1/2} \right).$$

That is, the correct stepsize to use is the reciprocal of the "preconditioned subspace smoothness constant".

To compute $\lambda_1 \left( (P_{\mathcal{B}} + \rho I)^{-1/2}(K_{\mathcal{B}\mathcal{B}} + \lambda I)(P_{\mathcal{B}} + \rho I)^{-1/2} \right)$, `GetStepsize` uses randomized powering [Kuczyński and Woźniakowski, 1992]. This technique has been used in several previous works on preconditioned optimization to great effect [Frangella et al., 2024a,b, Rathore et al., 2025]. Given a symmetric matrix $H$, preconditioner $P$, and damping $\rho$, randomized powering computes

$$\lambda_1((P + \rho I)^{-1/2}H(P + \rho I)^{-1/2}),$$

using only matrix-vector products with the matrices $H$ and $(P + \rho I)^{-1/2}$. When $P$ is calculated using `RandNysAppx`, `GetStepsize` can efficiently compute a matrix-vector product with $(P + \rho I)^{-1/2}$

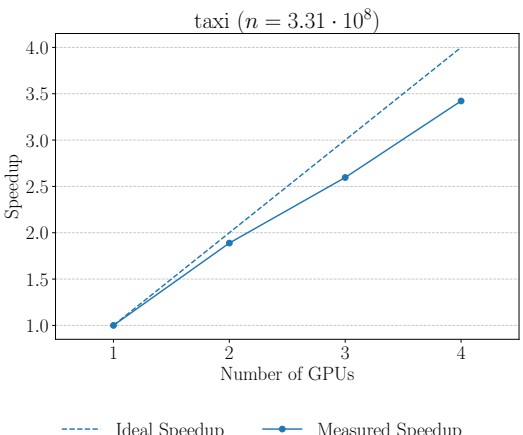

Figure 4: Multi-GPU scaling of `ADASAP` on the taxi dataset. `ADASAP` obtains near-linear scaling with the number of GPUs.

using (9). In practice, we find that 10 iterations of randomized powering are sufficient for estimating the preconditioned smoothness constant. The pseudocode for randomized powering, based on the presentation in Martinsson and Tropp [2020], is shown in Algorithm 6. Since `ADASAP` only runs `GetStepsize` for 10 iterations, the total cost of the procedure is $\mathcal{O}(b^2)$, which makes it the source of the $\mathcal{O}(b^2)$ cost of Phase III in Algorithm 2.

---

**Algorithm 6** `GetStepsize`

---

**Require:** symmetric matrix $H$, preconditioner $P$, damping $\rho$, maximum iterations $N \leftarrow 10$
    $v_0 \leftarrow \text{randn}(P.\text{shape}[0])$
    $v_0 \leftarrow v_0/\|v_0\|_2$                                                                              ▷ Normalize
    **for** $i = 0, 1, \ldots, N-1$ **do**
        $v_{i+1} \leftarrow (P + \rho I)^{-1/2} v_i$
        $v_{i+1} \leftarrow H v_{i+1}$
        $v_{i+1} \leftarrow (P + \rho I)^{-1/2} v_{i+1}$
        $v_{i+1} \leftarrow v_{i+1}/\|v_{i+1}\|_2$                                                              ▷ Normalize
    **end for**
    $\lambda \leftarrow (v_{N-1})^T v_N$
    **return** $1/\lambda$

---

### C.4 Nesterov acceleration

We present pseudocode for Nesterov acceleration in `NestAcc` (Algorithm 7).

---

**Algorithm 7** `NestAcc`

---

**Require:** iterates $W_t, V_t, Z_t$, search direction $D_t$, stepsize $\eta_{\mathcal{B}}$, acceleration parameters $\beta, \gamma, \alpha$
    $W_{t+1} \leftarrow Z_t - \eta_{\mathcal{B}} D_t$
    $V_{t+1} \leftarrow \beta V_t + (1-\beta)Z_t - \gamma \eta_{\mathcal{B}} D_t$
    $Z_{t+1} \leftarrow \alpha V_t + (1-\alpha)W_{t+1}$
    **return** $W_{t+1}, V_{t+1}, Z_{t+1}$

---

### C.5 Parallel scaling of `ADASAP`

Here we present Fig. 4, which shows the parallel scaling of `ADASAP` on the taxi dataset.

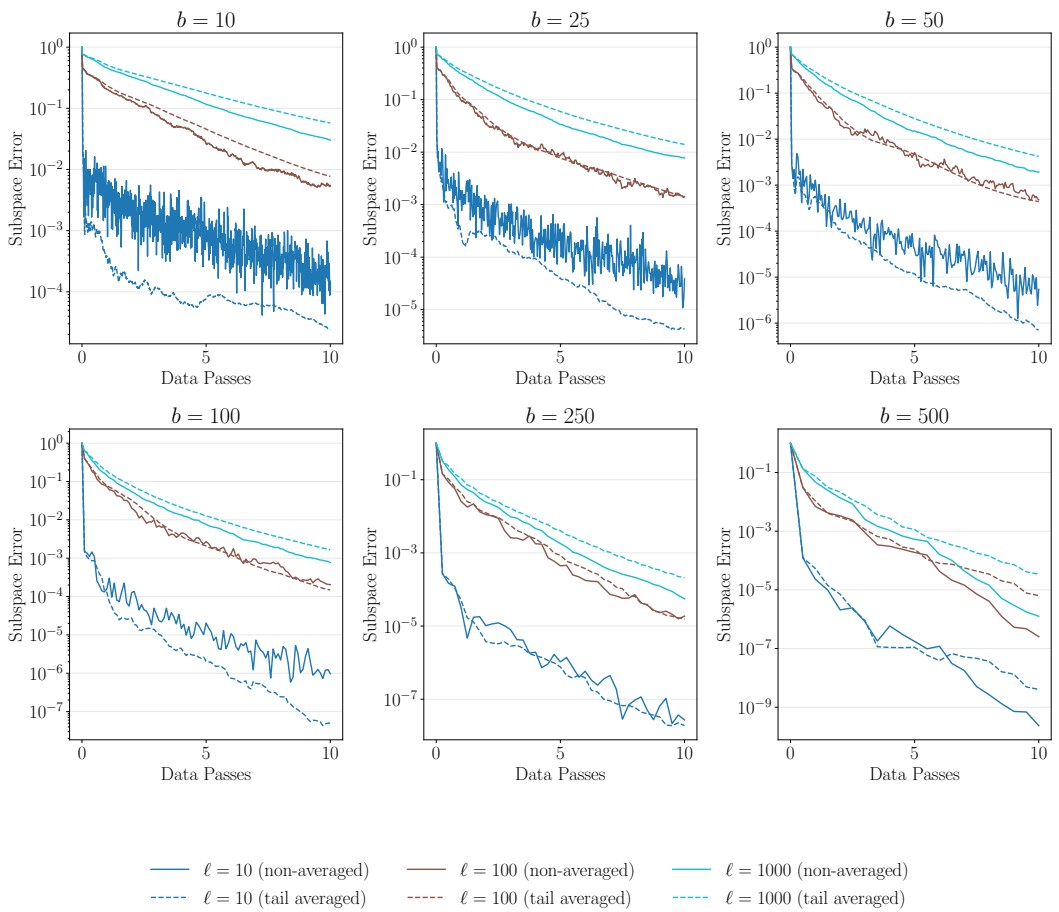

$$\ell = 10 \text{ (non-averaged)} \quad \ell = 100 \text{ (non-averaged)} \quad \ell = 1000 \text{ (non-averaged)}$$
$$\ell = 10 \text{ (tail averaged)} \quad \ell = 100 \text{ (tail averaged)} \quad \ell = 1000 \text{ (tail averaged)}$$

Figure 5: Performance of `ADASAP` with and without tail averaging. One "data pass" corresponds to one pass through the kernel matrix. Tail averaging does not improve convergence by a substantial margin.

### C.6  Impact of tail averaging on performance

The theoretical results in Section 3 require tail averaging. Here we assess whether tail averaging leads to practical improvements in the convergence of sketch-and-project algorithms for kernel ridge regression. To do so, we run a synthetic experiment using an RBF kernel with $n = 1000$ samples.

Fig. 5 displays the relative errors over the top-$\ell$ subspace, which are computed using the expression

$$\|\mathrm{proj}_\ell(\hat{m}) - \mathrm{proj}_\ell(m_n)\|_{\mathcal{H}}^2 / \|\mathrm{proj}_\ell(m_n)\|_{\mathcal{H}}^2.$$

When $\ell = 10$ and the blocksize $b$ is small, tail averaging slightly improves the convergence rate. However, when $\ell \in \{100, 1000\}$, tail averaging does not improve the convergence rate. In fact, as the blocksize increases, tail averaging results in *slower* convergence!

## D  Additional Details for Experiments

Here we provide additional details for the experiments that are not provided in the main paper.

### D.1  Determining hyperparameters for regression

We use a zero-mean prior for all datasets. We train the kernel variance, likelihood variance, and lengthscale (we use a separate lengthscale for each dimension of $X$) using the procedure of Lin et al. [2023], which we restate for completeness:

1. Select a centroid point from the training data $X$ uniformly at random.

2. Select the 10,000 points in the training data that are closest to the centroid in Euclidean norm.

3. Find hyperparameters by maximizing the exact GP likelihood over this subset of training points.

4. Repeat the previous three steps for 10 centroids and average the resulting hyperparameters.

### D.2 Optimizer hyperparameters

We present the hyperparameters for `ADASAP`, `ADASAP-I`, PCG, and SDD that were not described in the main paper.

For GP inference on large-scale datasets, we use blocksize $b = n/100$ in `ADASAP`, `ADASAP-I`, and SDD; blocksize $b = n/2{,}000$ for transporation data analysis, and $b = n/5$ for Bayesian optimization,

We set the rank $r = 100$ for both `ADASAP` and PCG.

Similar to Lin et al. [2024], we set the stepsize in SDD to be one of $\{1/n, 10/n, 100/n\}$ (this grid corresponds to SDD-1, SDD-10, and SDD-100), the momentum to 0.9, and the averaging parameter to $100/T_{\max}$.

### D.3 Additional details for GP inference experiments

song and houseelec are from the UCI repository, yolanda and acsincome are from OpenML, and benzene and malonaledehyde are from sGDML [Chmiela et al., 2017]. We select the kernel function $k$ for each dataset based on previous work [Lin et al., 2023, Epperly et al., 2024b, Rathore et al., 2025].

We standardize both the features and targets for each dataset. For fairness, we run all methods for an equal amount of *passes* through each dataset: we use 50 passes for yolanda, song, benzene, and malonaldehyde and 20 passes for acsincome and houseelec.

We use pathwise conditioning with 2,048 random features to (approximately) sample from the GP posterior.

### D.4 Additional details for transporation data analysis

We standardize both the features and targets and use a RBF kernel. Due to computational constraints, we use a single 99%-1% train-test split, and run each method for a single pass through the dataset.

### D.5 Additional details for Bayesian optimization

Our implementation of Bayesian optimization largely mirrors that of Lin et al. [2023]. We only present the high-level details here, and refer the reader to Lin et al. [2023] for the fine details of the implementation.

We draw the target functions $f : [0,1]^8 \to \mathbb{R}$ from a zero-mean GP prior with Matérn-3/2 kernel using 5,000 random features. At each iteration, we choose the acquisition points using parallel Thompson sampling [Hernández-Lobato et al., 2017]. As part of this process, we use the multi-start gradient optimization maximization strategy given in Lin et al. [2023]. At each step, we acquire 1,000 new points, which we use to evaluate the objective function. Concretely, if $x_{\text{new}}$ is an acquired point, we compute $y_{\text{new}} = f(x_{\text{new}}) + \epsilon$, where $\epsilon \sim \mathcal{N}(0, 10^{-6})$. We then add $(x_{\text{new}}, y_{\text{new}})$ to the training data for the next step of optimization. In our experiments, we initialize all methods with a dataset consisting of 250,000 observations sampled uniformly at random from $[0,1]^8$.

### D.6 Additional timing plots for Section 5.1

Here we present timing plots for the datasets used in Section 5.1 that were not shown in the main paper. Fig. 6 shows that all the methods (except PCG) perform similarly on both test RMSE and test mean NLL. However, Fig. 7 shows that `ADASAP` and PCG attain a much lower train RMSE than the competitors. This suggests that the similar performance of the methods on test RMSE and test

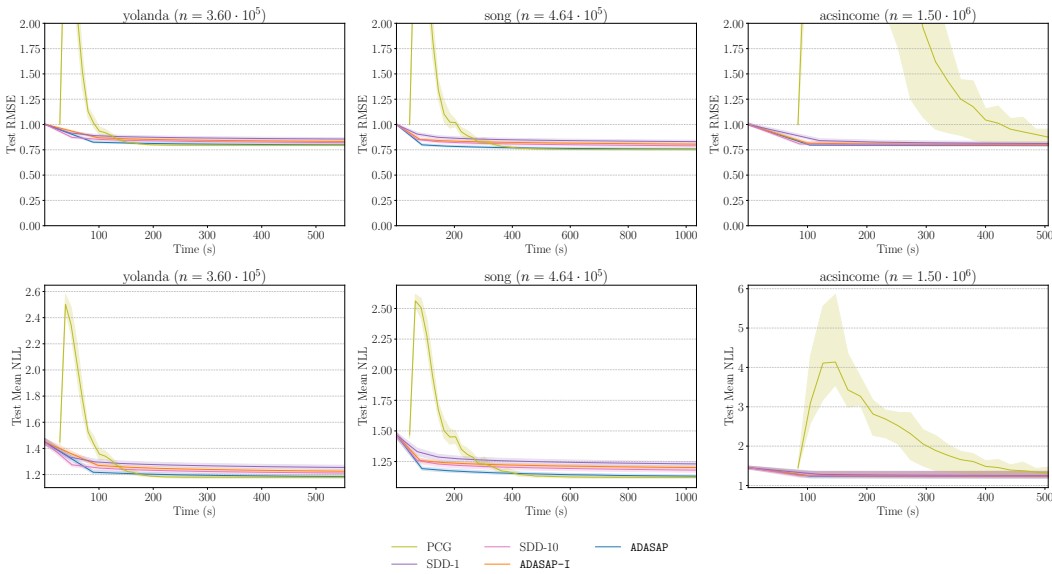

Figure 6: Performance of `ADASAP` and competitors on RMSE and mean NLL, as a function of time, for benzene, malonaldehyde, and houseelec. The solid curve indicates mean performance over random splits of the data; the shaded regions indicate the range between the worst and best performance over random splits of the data. `ADASAP` performs similar to the competition.

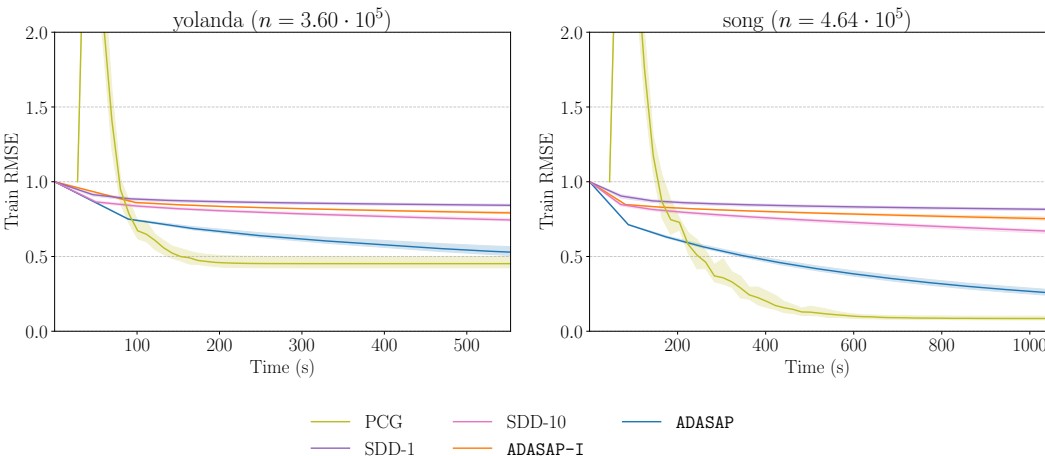

Figure 7: Performance of `ADASAP` and competitors on train RMSE, as a function of time, for yolanda and song. The solid curve indicates mean performance over random splits of the data; the shaded regions indicate the range between the worst and best performance over random splits of the data.

mean NLL is not due to differences in optimization, but rather, it is because the datasets are not well-modeled by Gaussian processes.

