# OpenReview forum: "Turbocharging Gaussian Process Inference with Approximate Sketch-and-Project"
_NeurIPS.cc/2025/Conference — NeurIPS 2025 poster_

### Official Review · Reviewer_5L3E · 2025-06-17

**Clarity:** 4
**Significance:** 3
**Originality:** 3
**Rating:** 5
**Confidence:** 3

**Summary:**

This paper proposes a scalable linear solver for Gaussian process (GP) posterior sampling, building on the sketch-and-project (SAP) framework of Gower and Richtárik (2015). The proposed method ADASAP integrates approximate subspace preconditioning via the randomized Nyström approximation (Tropp et al., 2017), distributed matrix-matrix products, and Nesterov acceleration. The authors also provide a convergence analysis of the SAP algorithm and evaluate ADASAP empirically on several large-scale regression datasets. Their results demonstrate the method's ability to scale to datasets with up to $10^8$ points, while showing competitive test RMSE and negative log-likelihood (NLL), and low variance with respect to different random data partitions.

**Questions:**

* How are RMSE and NLL computed?
* How are the RMSE and NLL of ADASAP influenced by input data dimensionality, block size, and tail averaging?

**Ethical Concerns:**

["NO or VERY MINOR ethics concerns only"]

**Final Justification:**

The authors have clarified my minor points during the discussion period. I maintain my score of 5 and recommend accepting the paper.

**Limitations:**

The authors are do not explicitly discuss limitations of their method. I do not see any obvious potential negative social impact.

**Paper Formatting Concerns:**

* Contributions 3 and 4 are not significantly different. Perhaps they could be merged into a single "Experimental Results" point.
* Line 72: The claim is inaccurate—conditioning requires cubic time in the number of data points.
* Line 135: The formulation “averaging the second half” is unclear; please clarify.

**Quality:**

3

**Strengths And Weaknesses:**

### **Quality**

The submission is technically sound and builds upon the recent GP inference literature. It combines existing components (SAP, randomized Nystrom approximations, distributed computation, and acceleration) in a coherent way, and the relation to prior work for GPs sampling is clearly explained. Although I am not familiar with the more theoretical results in the literature, the analysis of the error of the SAP algorithm seems like a meaningful contribution.

On the empirical side, the method performs well overall, particularly on the benzene, malonaldehyde, and houseelec datasets (Figure 2). However, the method’s relative performance is less convincing on the yolanda, song, and acsincome datasets (Appendix Figure 5). It would improve the presentation to move these results to the main text. Overall, the ADASAP the mean test RMSE and NLL seem to have low variance with respect to different random partitions of the data.

The presentation of experimental results in Table 1 is problematic. The use of mean values over 5 random partitions without standard errors makes it difficult to assess statistical significance even though Figures 2 and 5 suggest that this is not an issue. Furthermore, the policy of bolding results if they are within 0.01 of the best RMSE or NLL lacks rigor and should be changed with a more appropriate statistical comparison (e.g., paired sample t-tests) should be included.

Furthermore, some ablations that would enhance the empirical evaluation and improve the practical relevance of the work are currently missing, including the influence on RMSE and NLL of:
* Input feature dimensions (most of the datasets in the evaluation have $< 15$ input dimensions);
* Block size;
* Tail averaging.
These would help clarify when the method is most effective.

### **Clarity**

The paper is well written and enjoyable to read. The structure and the scope of the work are clear. Connections to related work are also well presented. Code is provided though I did not executed it. Nevertheless, there are areas where clarity could be improved:

* A more formal introduction to the SAP framework would benefit readers unfamiliar with this class of methods.
* Important implementation details (e.g., computation of RMSE and mean NLL) are not reported.
* More algorithms should be moved to the main text to make the paper more complete.

### **Significance**

The work addresses the important challenge of scaling GP posterior sampling. While the algorithmic contribution is somewhat incremental, the proposed method scales to large datasets and shows good performance. The convergence analysis adds some theoretical depth.

However, the claim that this is "the first time that full GP inference has been scaled to a dataset of this size" (Line 92) is overstated. As far as I understand, the large-scale experiment on the TAXI dataset ($10^8$) only computes the posterior mean (which is equivalent to the kernel ridge regression estimator) for computational budget reasons. Similar scaling has been achieved previously for the kernel ridge regression estimator, e.g., in [1], on an even larger partitioning of the TAXI dataset which exceeds $10^9$ examples. The authors should acknowledge this work, consider comparing with this method, and clarify the novelty of their approach more carefully.

Furthermore, the paper does not include an ablation study of the effect of the approximate preconditioner and acceleration which would improve the paper's significance.

[1] Kernel methods through the roof: handling billions of points eﬃciently, Meanti et al., 2020

### **Originality**

The paper combines well-established components in a practical and effective way. The improved SAP for posterior sampling in GPs, along with a convergence analysis in RKHS norm, constitutes an incremental but meaningful contribution.

### **Recommendations**

Overall, I recommend accepting this work. To strengthen the paper, I suggest the following:

* Improve statistical reporting in Table 1 (add standard errors, t-tests);
* Add more algorithms to the main text;
* Include the results in Appendix Figure 5 to the main text;
* Clarify computation for RMSE and NLL;
* Introduce the SAP method more formally;
* Consider adding ablations on input dimensionality, block size, and tail averaging;
* Moderate the claims about scalability and acknowledge relevant prior work more precisely.

---

> ### Author Rebuttal · Authors · 2025-07-30
>
> > **Comment 4.1** — The presentation of experimental results in Table 1 is problematic. The use of
> mean values over 5 random partitions without standard errors makes it difficult to assess statistical
> significance even though Figures 2 and 5 suggest that this is not an issue. Furthermore, the policy of
> bolding results if they are within 0.01 of the best RMSE or NLL lacks rigor and should be changed
> with a more appropriate statistical comparison (e.g., paired sample t-tests) should be included.
>
> We appreciate your suggestions for improvement.
> In the revision we will provide standard errors, and we will use a paired t-test for comparing the methods in Table 1.
>
> > **Comment 4.2** — Furthermore, some ablations that would enhance the empirical evaluation and improve the practical relevance of the work are currently missing, including the influence on RMSE and NLL of:
> > * Input feature dimensions (most of the datasets in the evaluation have $< 15$ input dimensions);
> > * Block size;
> > * Tail averaging. These would help clarify when the method is most effective.
>
> * The dimensions of the datasets are: yolanda $(d = 100)$, song $(d=90)$ benzene $(d = 66)$, malonaldehyde $(d=36)$, acsincome $(d=9)$, houseelec $(d=9)$, and taxi $(d=9)$, so the majority of the inputs do not have feature dimensions $< 15$.
> We will be sure to include this information in the revision.
>
> * The theory developed in our paper shows that the convergence rate improves as the blocksize increases.
> However, the improvement in convergence rate must be balanced with increased computational and memory costs.
> This phenomenon has been verified in many prior SAP papers.
> Hence, for brevity we have not studied this here, but we will point to the references that show this in the revision.
> In practice, we recommend using the blocksize that maximizes GPU utilization while not running out of memory.
>
> * As discussed in Section 4.2.1, we believe the value of tail averaging is primarily theoretical, as has been observed in the SGD literature (Shamir and Zhang, 2013; Johnson and Zhang, 2013), where the last iterate often delivers comparable performance to the averaged iterate.
> In the revision, we will add experiments to show the impact of tail averaging on convergence.
>
> > **Comment 4.3** — Nevertheless, there are areas where clarity could be improved:
> > * A more formal introduction to the SAP framework would benefit readers unfamiliar with this class of methods.
> > * Important implementation details (e.g., computation of RMSE and mean NLL) are not reported.
> > * More algorithms should be moved to the main text to make the paper more complete.
>
> * Section 2 provides a brief introduction to SAP and its properties most relevant for our setting.
> We would like to have provided more intuition for SAP, but could not owing to length constraints.
> Finally, section A.2 provides a detailed literature review of SAP methods, from which unfamilar readers can learn more.
>
> * Our computation of these quantities follows Lin et al. (2023; 2024).
> The provided codebase also provides all the code used to compute these quantities in our experiments.
> The precise mathematical formulae we used are provided in response to Comment 4.6.
>
> * Unfortunately, due to length constraints were forced to push the description of many algorithmic subroutines to the appendix.
> We agree this is not necessarily ideal.
> If accepted, we get an extra page, and will try and bring more of these algorithms forward to the main paper.
>
> > **Comment 4.4** — However, the claim that this is "the first time that full GP inference has been scaled to a dataset of this size" (Line 92) is overstated. As far as I understand, the large-scale experiment on the TAXI dataset ($10^8$) only computes the posterior mean (which is equivalent to the kernel ridge regression estimator) for computational budget reasons. Similar scaling has been achieved previously for the kernel ridge regression estimator, e.g., in [1], on an even larger partitioning of the TAXI dataset which exceeds $10^9$ examples. The authors should acknowledge this work, consider comparing with this method, and clarify the novelty of their approach more carefully.
>
> When we refer to full GP inference, we mean performing GP inference on the entire dataset, without using inducing points.
> While Falkon in [1] runs on $10^{9}$ samples, they use $10^{5}$ inducing points, which means they are processing a kernel matrix of size $10^5 \times 10^9$.
> On the other hand, the kernel matrix that we process has size $3.31 \cdot 10^8 \times 3.31 \cdot 10^8$---which is three orders of magnitude larger than the kernel matrix processed by Falkon.
> Furthermore, inducing points methods for GPs and kernel ridge regression often perform worse than methods that solve the full problem (Gardner et al., 2018; Lin et al., 2023; 2024; Rathore et al., 2025).
> Therefore, we do not think it is necessary to perform a comparison with Falkon at this time.
>
> > **Comment 4.5** — Furthermore, the paper does not include an ablation study of the effect of the approximate preconditioner and acceleration which would improve the paper's significance.
>
> * Approximate preconditioner: Incorporating the Nyström approximation allows ADASAP to scale to larger blocksizes.
> We did not run ADASAP without the Nyström approximation, since it would not scale to our largest datasets.
> For example, on taxi $(n = 3.31 \cdot 10^8)$, we run ADASAP with blocksize $b = 1.65 \cdot 10^5$.
> Computing the Cholesky decomposition to solve the $b \times b$ linear system with $K_{\mathcal B \mathcal B} + \lambda I$ would require $\mathcal O(b^3)$ compute and $\mathcal O(b^2)$ memory, which is not practical for this blocksize.
> Since the "full" preconditioner does not scale to this blocksize, we omitted it from our experiments.
>
> * Acceleration: Several works have shown that incorporating acceleration into KRR solution methods leads to consistent practical improvements (Tu et al., 2017; Rathore et al., 2025).
> Therefore, in the interest of time, we did not run experiments without acceleration.
>
> > **Comment 4.6** — How are RMSE and NLL computed?
>
> Let the test set be $\lbrace (x_i, y_i) \rbrace_{i = 1}^{n_{\mathrm{tst}}}$.
> Furthermore, let $m_n(\cdot)$ be the mean of the GP posterior.
>
> We compute the RMSE using the expression $\sqrt{\frac{1}{n_{\mathrm{tst}}} \sum_{i = 1}^{n_{\mathrm{tst}}}(m_n(x_i) - y_i)^2}$.
>
> For the mean NLL, we compute 64 samples from the posterior GP using pathwise conditioning, and we use those samples to compute empirical variance estimates $\sigma_i^2$ for each point $i$ in the test set.
> We then compute the negative log-likelihood for each point $i$ in the test set using $\mathcal N(m_n(x_i), \sigma_i^2)$ as the "reference" distribution.
> Finally, we take the mean of these negative log-likelihoods to get the mean NLL.
>
> Our approach for computing RMSE and mean NLL is identical to that of Lin et al. (2023; 2024).
> Please see our code for more details.
>
> > **Comment 4.7** — How are the RMSE and NLL of ADASAP influenced by input data dimensionality, block size, and tail averaging?
>
> Please see our response to Comment 4.2.
>
> > **Comment 4.8** — Contributions 3 and 4 are not significantly different. Perhaps they could be merged into a single "Experimental Results" point.
>
> We originally separated these contributions because they were addressing different experimental settings (GPs and Bayesian optimization) which have different use cases.
> However, we will consider merging these in the revision to improve clarity.
>
> > **Comment 4.9** — Line 72: The claim is inaccurate—conditioning requires cubic time in the number of data points.
>
> We are confused by the point the reviewer is making---in general, computing the square root of a $n \times n$ matrix requires $\mathcal O(n^3)$ time.
> Please let us know if we have misinterpreted your point.
>
> > **Comment 4.10** — Line 135: The formulation “averaging the second half” is unclear; please clarify.
>
> When we say ``averaging the second half'' we are referring to the tail averaging in Algorithm 1 (see the if statement in the algorithm block).
> We will make sure that this is clarified in the revision.
>
> **References**
>
> Jacob Gardner, Geoff Pleiss, Kilian Q Weinberger, David Bindel, and Andrew G Wilson. GPyTorch:
> Blackbox matrix-matrix Gaussian process inference with GPU acceleration. In Advances in
> Neural Information Processing Systems, 2018.
>
> Rie Johnson and Tong Zhang. Accelerating stochastic gradient descent using predictive variance
> reduction. In Advances in Neural Information Processing Systems, 2013.
>
> Jihao Andreas Lin, Javier Antoran, Shreyas Padhy, David Janz, José Miguel Hernández-Lobato,
> and Alexander Terenin. Sampling from Gaussian process posteriors using stochastic gradient
> descent. In Advances in Neural Information Processing Systems, 2023.
>
> Jihao Andreas Lin, Shreyas Padhy, Javier Antoran, Austin Tripp, Alexander Terenin, Csaba Szepesvari, José Miguel Hernández-Lobato, and David Janz. Stochastic gradient descent for Gaussian
> processes done right. In International Conference on Learning Representations, 2024.
>
> Pratik Rathore, Zachary Frangella, Jiaming Yang, Michal Dereziński, and Madeleine Udell. Have
> ASkotch: A neat solution for large-scale kernel ridge regression. arXiv preprint 2407.10070, 2025.
>
> Ohad Shamir and Tong Zhang. Stochastic gradient descent for non-smooth optimization: Convergence results and optimal averaging schemes. In Proceedings of the 30th International Conference on Machine Learning, 2013.
>
> Stephen Tu, Shivaram Venkataraman, Ashia C. Wilson, Alex Gittens, Michael I. Jordan, and
> Benjamin Recht. Breaking locality accelerates block Gauss-Seidel. In Proceedings of the 34th
> International Conference on Machine Learning, 2017.

---

> > ### Comment · Reviewer_5L3E · 2025-08-01
> >
> > Thank you for your clarifications and answers to my questions. I maintain my score.

---

### Official Review · Reviewer_WCDC · 2025-06-27

**Clarity:** 3
**Significance:** 3
**Originality:** 3
**Rating:** 4
**Confidence:** 5

**Summary:**

The paper proposes a GP inference algorithm based on sketch and project (a sort of sketched Newton method) which is then combined with approximate preconditioning, distributed operations, and Nesterov acceleration. It derives guarantees (specifically, for posterior mean estimation) for the sketch and project algorithm, characterizing the convergence rate (Theorem 3.2) and time complexity (Corollary 3.3) as a function of the spectrum of the GP kernel matrix. These results show transitions from sublinear to linear rates and show dependence (or lack thereof) on a reduced (smoothed) condition number.

**Questions:**

See weaknesses

**Ethical Concerns:**

["NO or VERY MINOR ethics concerns only"]

**Final Justification:**

I do not dissent from the consensus of accepting the paper, but I do feel that there are substantial changes in the claims that are required for the paper to fairly present its contributions and place them in the existing literature. I have kept my score at 4 because this is the stage at which I believe the current manuscript is. But if this entails rejecting the paper, then I'm happy to raise my score to 5.

**Limitations:**

Yes

**Quality:**

3

**Strengths And Weaknesses:**

## Strengths

The proposed algorithm lies between (preconditioned) coordinate descent and stochastic gradient descent methods, trading off stability and the handling of ill-conditioning. The theoretical results give an informative characterization of the convergence and time complexity of SAP for the mean path of GP posteriors. Results parallel those in the setting of kernel ridge regression developed by Rathore et al. The theory is supported by numerical evaluations comparing two exact solver baselines (PCG and SDD) on 3 tasks. The paper is accompanied by a well-structured and clean codebase.


## Weaknesses :

- The related work in Section A.2 does not substantially compare the results in Theorem 3.2 and Corollary 3.3 to existing ones. In particular, the result in Theorem 3.2 appears to closely parallel those for ASkotch from Rathore et al. but with a smoothed conditioned number. Considering the close relation between KRR and the GP mean, a proper comparison is warranted.

- The "condition number-free" results do not appear to be particularly relevant, given that it only applies for large enough $\epsilon$. In other words, the algorithm is conditioned number-free for large error compared to the kernel matrix conditioning. These do not seem to be situations that are meaningful in practice: if $K$ is severely ill-conditioned, we could really only expect to recover $m$ along very few dimension $\ell \ll n$ using a larger $\lambda$ to regularize the problem. Otherwise, the second term will end up dominating the first one for any reasonable choice of $\epsilon$. Hence, the "condition number-free" regime does not appear to be of practical interest (except perhaps in a very narrow setting). Claims of "condition number-free" convergence throughout the paper should be put in significantly more context or be toned down (particularly in the abstract and introduction, e.g., lines 10 and 49).

- On line 39, the manuscript states that ADASAP "obtains the robustness to ill-conditioning of PCG". I imagine that the paper alludes here to the dependence on the smoothed condition number $\phi$, which remains a condition number. What is more, even in the benign setting of Corrolary 3.3, any robustness gains compared to PCG appear to be due to the restriction of the estimation to the principal $\ell$ components of the kernel matrix. As such, there is an additional "approximation error" to take into account, namely, $\|m_n - proj_\ell(m_n)\|$ which is on the order of the tail of the spectrum of $K$. As such, the robustness gains also comes at the cost of a larger approximation error. This point (which should be made significantly clearer in the manuscript) suggest the importance of including an experimental investigation to further support the robustness claims (or additional theoretical results showing that equilibrating approximation and estimation error is not harmful).

- While the paper is well-written and clear, addressing a few minor issues could improve the presentation:
   - Some notation is not defined: $n_{rhs}$ (line 110), $K_{\lambda}$ (Theorem 3.2), $\rho$ (Line 199), $\mathcal{Y}_{\mathcal{B}}$ (Line 106)
   - The suffix for SDD-10, SDD-100 etc. is only explained in the appendix (Line 927). This should have been explained before or in the caption of Figure 1.
   - The significance of tail averaging is only discussed in Sec 4.2.1.

---

> ### Author Rebuttal · Authors · 2025-07-30
>
> > **Comment 3.1** — The related work in Section A.2 does not substantially compare the results in Theorem 3.2 and Corollary 3.3 to existing ones. In particular, the result in Theorem 3.2 appears to closely parallel those for ASkotch from Rathore et al. but with a smoothed conditioned number. Considering the close relation between KRR and the GP mean, a proper comparison is warranted.
>
> We are happy to discuss how the theoretical results we have developed compare to those for ASkotch in Rathore et al.
> Our analysis in Section 3 focuses on the convergence of the posterior mean along the top $\ell$-subspace using exact SAP with tail-averaging.
> Our motivation for considering this is prior works such as
>  Dicker et al. (2017); Lin et al. (2020; 2023), which show theoretically and empirically that a low-dimensional top eigenspace of $K + \lambda I$ is most relevant for achieving good generalization.
>
> In contrast, Rathore et al. focuses on convergence of approximate sketch-and-project (with and without acceleration) *over the entire space*, and as a result, they are only able to show a fast convergence guarantee for ASkotch under certain strong conditions on the spectral decay (effectively limiting the condition number of the problem).
> Consequently, the global convergence analysis of Rathore et al. does not explain the rapid initial progress that SAP-style algorithms make on test error.
> We believe our two-phase analysis consisting of (i) fast condition number-free sublinear convergence, followed by (ii) a slower global convergence rate dependent upon the subspace condition number, better captures this phenomenon.
> In particular, our result is the first step in developing a systematic analysis of the generalization properties of SAP-style algorithms.
> In the future, we would like to provide a systematic analysis of the convergence and generalization properties of approximate SAP methods by combining our approach in this work with that of Rathore at al.
>
> In the revision, we shall add the preceding discussion on how our results compare to those of Rathore et al. to the related work section.
>
> > **Comment 3.2** — The "condition number-free" results do not appear to be particularly relevant, given that it only applies for large enough $\epsilon$. In other words, the algorithm is conditioned number-free for large error compared to the kernel matrix conditioning. These do not seem to be situations that are meaningful in practice: if $K$ is severely ill-conditioned, we could really only expect to recover along very few dimension $\ell \ll n$ using a larger $\lambda$ to regularize the problem. Otherwise, the second term will end up dominating the first one for any reasonable choice of $\epsilon$. Hence, the "condition number-free" regime does not appear to be of practical interest (except perhaps in a very narrow setting). Claims of "condition number-free" convergence throughout the paper should be put in significantly more context or be toned down (particularly in the abstract and introduction, e.g., lines 10 and 49).
>
> We agree that the condition number-free sublinear convergence in Theorem 3.2 and Corollary 3.3 only applies up to a certain moderate precision (large enough $\epsilon$), whereas for high precision (smaller $\epsilon$), the condition number-dependent linear rate takes over.
> However, arguably, the moderate precision regime (large enough $\epsilon$) is particularly relevant in the context of GP as well as other machine learning settings, where the ultimate goal is to optimize the test error, as opposed to the training error.
> Once the algorithm reaches a certain moderate precision in the training error, it has typically achieved close to optimal test error and further optimization yields little further benefit.
> In fact, the phase-one sublinear convergence rate appears to align more closely with the behavior of the test error in our empirical results than the second-phase linear rate does.
>
> That being said, we accept that our condition number-free sublinear rate is relevant only for a range of $\epsilon$ values.
> We will provide additional context to make this clear in the paper, particularly in the introduction.
>
> > **Comment 3.3** — On line 39, the manuscript states that ADASAP "obtains the robustness to ill-conditioning of PCG". I imagine that the paper alludes here to the dependence on the smoothed condition number $\phi$, which remains a condition number. What is more, even in the benign setting of Corrolary 3.3, any robustness gains compared to PCG appear to be due to the restriction of the estimation to the principal $\ell$ components of the kernel matrix. As such, there is an additional "approximation error" to take into account, namely, $|m_n - proj_{\ell}(m_n)|$ which is on the order of the tail of the spectrum of $K$.
> As such, the robustness gains also comes at the cost of a larger approximation error. This point (which should be made significantly clearer in the manuscript) suggest the importance of including an experimental investigation to further support the robustness claims (or additional theoretical results showing that equilibrating approximation and estimation error is not harmful)
>
> As mentioned above, prior works such as Dicker et al. (2017); Lin et al. (2020; 2023) provide both theoretical and empirical evidence that a low-dimensional top eigenspace of $K + \lambda I$ is most relevant for achieving good generalization.
> In other words, the error restricted to the top principal components tends to align better with the test error, whereas the remaining "tail" approximation error tends to be associated with overfitting.
> While these statements are informal (and also, depend on the choice of $\ell$), the robustness to ill-conditioning is observed empirically in our results, given the strong test error convergence of ADASAP.
> We also note that one can freely set $\ell=n$ both in Theorem 3.2 and in Corollary 3.3, which brings the tail approximation error to 0 and still leads to a convergence guarantee.
>
> Nevertheless, we fully agree with the reviewer's point about clarifying the meaning of robustness.
> In particular, we will discuss the role of the approximation error along the tail principal components, and how this affects the robustness of our algorithms.
>
> > **Comment 3.4** — While the paper is well-written and clear, addressing a few minor issues could improve the presentation:
> > * Some notation is not defined: $n_{\mathrm{rhs}}$ (line 110), $K_\lambda$ (Theorem 3.2), $\rho$ (Line 199),  $\mathcal Y_B$ (Line 106)
> > * The suffix for SDD-10, SDD-100 etc. is only explained in the appendix (Line 927). This should have been explained before or in the caption of Figure 1.
> > * The significance of tail averaging is only discussed in Sec 4.2.1.
>
> * Thanks for pointing out the undefined notation---we will fix this in the revision.
>
> * We also explain the suffix around line 240 in the main submission.
> However, we agree that explaining the suffix in the caption of Figure 1 would improve clarity, and we will update this in the revision.
>
> * We also provide a high-level overview of why we use tail averaging in section 3.1 (see lines 108-109).
> Due to limited space, we could not elaborate further in the main submission---the details of the analysis are in Appendix B.
> If there are particular details you would like us to add to the main submission, please let us know.
>
> **References**
>
> Lee H Dicker, Dean P Foster, and Daniel Hsu. Kernel ridge vs. principal component regression: Minimax bounds and the qualification of regularization operators. Electronic Journal of Statistics, 11:1022–1047, 2017.
>
> Jihao Andreas Lin, Javier Antoran, Shreyas Padhy, David Janz, José Miguel Hernández-Lobato, and Alexander Terenin. Sampling from Gaussian process posteriors using stochastic gradient descent. In Advances in Neural Information Processing Systems, 2023.
>
> Junhong Lin, Alessandro Rudi, Lorenzo Rosasco, and Volkan Cevher. Optimal rates for spectral algorithms with least-squares regression over Hilbert spaces. Applied and Computational Harmonic Analysis, 48(3):868–890, 2020.

---

> > ### Comment · Reviewer_WCDC · 2025-08-04
> >
> > I thank the authors for tackling on my concerns.
> >
> > **Comment 3.1**: I trust that the authors will take care to include this discussion prominently in the paper as it more adequately puts their work in context of prior literature.
> >
> > **Comment 3.2**: The authors' claim that "Once the algorithm reaches a certain moderate precision in the training error, it has typically achieved close to optimal test error" is not substantiated and it is certainly not the case in ML settings: given the appropriate "inductive bias" (i.e., appropriate kernel), interpolating solutions provide optimal test errors in a myriad of settings. It is therefore certainly not the case that $\epsilon$ should be kept "moderately large." Even if that were the case, what constitutes "moderate" for generalization error need not be large enough to reach the "condition number-free" regime. At least the manuscript provides no theoretical or empirical validation of this claim.
> >
> > I strongly advise that the manuscript be revised in its claims of "condition number-free sublinear rate" throughout and not just in the introduction. As it is, there is no evidence in the work supporting the fact that this property holds for the method in any setting of interest.
> >
> > **Comment 3.3**: As above, note that "the error restricted to the top principal components tends to align better with the test error" is heavily dependent on the spectral properties of $K$, particularly when regularization is involved. Also, setting $\ell = n$ as suggested would negate many of the smoothed condition number and robustness advantages of the currently proposed method. This issue is admittedly less critical than the previous one as regularization and properties of $K$ often mean that smaller $\ell$ are of interest. Regardless, more careful discussion of these trade-offs (e.g., robustness vs accuracy) should be included.

---

### Official Review · Reviewer_UCmX · 2025-07-02

**Clarity:** 2
**Significance:** 4
**Originality:** 3
**Rating:** 5
**Confidence:** 4

**Summary:**

The paper provides a new optimization method for Gaussian Processes that is very scalable.  It builds on the Sketch-and-Project (SAP) framework that uses a Nystrom sampling sketch at each step of the optimization.  The advantage of the new approach is better dependence on batch size b, by using a numerically stable method to update the estimation of the sampled gram matrix K.  The procedure also uses Nesterov's acceleration and can be parallelized.

The paper provides some analysis which shows the convergence depends on the stable rank of the gram matrix, and with sub-quadratic update time if K exhibits polynomial spectral decay.
The experiments use cross-validation on large datasets to validate the method.  They show comparison to a couple leading methods from the past couple years, and show better fit.  They also scale to a NYC Taxi dataset with over 30 million data points -- where other methods fail.

**Questions:**

I would like to see more ablation studies to explain which of the improvements on any baselines and the basic SAP make the biggest contribution in the scaling.  Is this possible to do?  If not, please try to give insight into which components are most essential in the improvements.

**Ethical Concerns:**

["NO or VERY MINOR ethics concerns only"]

**Final Justification:**

This is a real solid contribution in that it provides what is now the state of the art way to scale up a widely used approach: Gaussian Processes.  I think the experimental section could be better, but the paper explain how it works well, and the results are clear important advancements, so I think it should be accepted.

**Limitations:**

Yes.

**Quality:**

4

**Strengths And Weaknesses:**

There is room for improvement in the writing.  The algorithm description came after the section describing the convergence results.  So it was fairly difficult to understand the paper without going back and forth several times.

Also, there seems to be a few extensions to the basic SAP framework.  The faster update, the acceleration, and the parallelization.  While it was useful to see the ADASAP-I that did not do preconditioning, it would be useful to see the effect of each of those three steps (the A, D, and A) in separate runs of the algorithm so we can see which is responsible for the increased improvement.

Gaussian Processes are a core technique in ML, and scaling them has been a challenge.  This paper while not centered around a single grand idea, does provide a very scalable solution for GP, and I think that make it a strong contribution.

---

> ### Author Rebuttal · Authors · 2025-07-30
>
> > **Comment 2.1** — There is room for improvement in the writing. The algorithm description came after the section describing the convergence results. So it was fairly difficult to understand the paper without going back and forth several times.
>
> We appreciate hearing about your experience as a reader, and will add more guideposts and references in the revision to enable smoother transitions between sections.
> We structured the paper so that results on each algorithm (SAP, ADASAP) appear in the section introducing that algorithm.
> The attractions and limitations of SAP are what motivate the development of ADASAP, which we introduce in Section 4.
>
> > **Comment 2.2** — Also, there seems to be a few extensions to the basic SAP framework. The faster update, the acceleration, and the parallelization. While it was useful to see the ADASAP-I that did not do preconditioning, it would be useful to see the effect of each of those three steps (the A, D, and A) in separate runs of the algorithm so we can see which is responsible for the increased improvement.
>
> Here is how we view the impacts of the extensions to the basic SAP framework:
>
> * Faster update: Incorporating the Nyström approximation allows ADASAP to scale to larger blocksizes.
> We did not run ADASAP without the Nyström approximation, since it would not scale to our largest datasets.
> For example, on taxi $(n = 3.31 \cdot 10^8)$, we run ADASAP with blocksize $b = 1.65 \cdot 10^5$.
> Computing the Cholesky decomposition to solve the $b \times b$ linear system with $K_{\mathcal B \mathcal B} + \lambda I$ would require $\mathcal O(b^3)$ compute and $\mathcal O(b^2)$ memory, which is not practical for this blocksize. In terms of wall-clock time, computing the Cholesky decomposition in this setting would probably correspond to over a day on an A100 GPU.
>
> * Acceleration: Several works have shown that incorporating acceleration into KRR solution methods leads to consistent practical improvements  (Tu et al., 2017; Rathore et al., 2025).
> Therefore, in the interest of time, we did not run experiments without acceleration.
>
> * Parallelization: We show the impact of parallelization on the taxi dataset in Figure 4 of the Appendix.
> ADASAP achieves a speedup of 3.4$\times$ when using 4 GPUs, demonstrating that our method benefits from parallelization.
>
> > **Comment 2.3** — I would like to see more ablation studies to explain which of the improvements on any baselines and the basic SAP make the biggest contribution in the scaling. Is this possible to do? If not, please try to give insight into which components are most essential in the improvements.
>
> Please see the response to comment 2.2.
> We believe that the biggest contribution in scaling is due to the faster update (which allows for larger blocksizes), followed by acceleration (helps for ill-conditioned problems like KRR), followed by parallelization.
> For smaller problems, we would not expect parallelization to be as helpful, since small problems may not fully utilize all of the GPUs.
>
> **References**
>
> Pratik Rathore, Zachary Frangella, Jiaming Yang, Michal Dereziński, and Madeleine Udell. Have
> ASkotch: A neat solution for large-scale kernel ridge regression. arXiv preprint 2407.10070, 2025.
>
> Stephen Tu, Shivaram Venkataraman, Ashia C. Wilson, Alex Gittens, Michael I. Jordan, and
> Benjamin Recht. Breaking locality accelerates block Gauss-Seidel. In Proceedings of the 34th
> International Conference on Machine Learning, 2017.

---

> > ### Comment · Reviewer_UCmX · 2025-08-03
> >
> > I understand that not parallelized/distributed baselines are implicit.  However, the rational for not breaking down the effect of Nystrom approximation and Acceleration are not very satisfying.
> >
> > If the claims are that Acceleration are known to be effective, then they should be more strong incorporated into baselines.  This may suggest that it should be implicit, and not in the name of your method.
> >   Although I think just running your method without acceleration and showing the effective would be more useful and appropriate.
> >
> > You can demonstrate the effect of Nystrom approximation on a smaller dataset, or on synthetic data.  I understand this is also a well-studied trick, but within the context of the size and accuracy of your proposed approach ADASAP, it would still be useful to understand what the trade-offs are.  For instance, at what scale should one use "DASAP" (without Nystrom approximation) instead of ADASAP?  This useful aspect is not explained well in the paper.
> >
> > The paper is already a nice contribution, but it would be more scholarly with these extra experiments.

---

### Official Review · Reviewer_P6N1 · 2025-07-02

**Clarity:** 3
**Significance:** 2
**Originality:** 4
**Rating:** 5
**Confidence:** 3

**Summary:**

The authors propose a new algorithm for Gaussian Process (GP) inference, with a focus on addressing large-scale problems. They identify two key issues in previous methods (PCG and SDD): robustness to ill-conditioning and scalability. Their method addresses both, offering good default hyperparameters and distributed execution capability. The paper also provides convergence guarantees for the posterior approximation under ill-conditioning (applicable more generally to sketch-and-project methods), and demonstrates superior performance on a series of benchmark tasks.

**Questions:**

1) When the dataset becomes very large (as in the main scenario considered in the paper), in my experience, it often makes sense to move away from GPs and use alternative models like Bayesian Neural Networks or Deep Kernel Methods, which don’t suffer from the same scaling issues. Could the authors elaborate on why GPs remain a preferable choice in this context? Some comparative discussion would strengthen the case for GPs.

2) Existing state-of-the-art GP methods already handle datasets on the order of 10^6. So, could authors provide any details on how much additional value is gained by pushing beyond this scale? This ties into the previous question about the motivation for continuing with GPs at such scales.

3) I didn’t fully grasp two points that might be critical. Could the authors clarify what they mean by “condition number-free” and what exactly is considered “ill-conditioning”? I assume this refers to challenges in inverting the matrix K + \lambda I, but I’m not entirely certain that’s the intended meaning.

4) In Algorithm 1, the variable ‘b’ appears without an immediate explanation. While it becomes clearer later in Section 3.2, it would help to define it properly when it first appears, especially since it plays a role in justifying the SAP algorithm’s reduced cost. Also, practical guidance on how to choose the block size ‘b’ would be very helpful.

5) Could the authors provide references or direct links to the proofs of Theorem 3.2 and Corollary 3.3 in the Appendix when they are first introduced? Currently, it’s a bit inconvenient to search the paper and then the Appendix for these proofs.

**Ethical Concerns:**

["NO or VERY MINOR ethics concerns only"]

**Final Justification:**

The paper presents a well-motivated and technically solid contribution to scalable GP inference, with strong empirical results and theoretical backing. While further evaluation of the uncertainty estimates would strengthen the work, the proposed method addresses important limitations of prior approaches and represents a meaningful advancement.

**Limitations:**

The authors acknowledge certain limitations, such as not covering approximate preconditioning, acceleration, or uniform sampling in their theoretical analysis, which is appreciated.

However, the discussion of limitations and potential future work could be expanded. For example, what kinds of problems are particularly challenging for the method? How do the authors recommend practitioners use the method in different contexts? It would be helpful to draw comparisons to non-GP alternatives.

While the authors emphasize that the algorithm performs well with default hyperparameters (which is great for adoption), more discussion around what might go wrong and how one should adjust the hyperparameters accordingly would be useful for practitioners.

Lastly, should the term “large-scale” be explicitly mentioned in the paper’s title? It could help clarify the paper’s focus and improve visibility beyond the GP research community.

**Paper Formatting Concerns:**

No concerns.

**Quality:**

4

**Strengths And Weaknesses:**

Strengths:

1) The proposed algorithm performs well, and its ability to scale to datasets with more than 3 x 10^8 observations is particularly impressive.

2) The paper establishes convergence of the posterior mean, and the theoretical contributions appear sound (though I have some follow-up questions in the respective section).

3) The distributed nature of the algorithm is a practical strength that could boost its adoption.

4) The paper is well-written and, to the best of my knowledge, presents original work.

Weaknesses:

1) The justification for the algorithm primarily focuses on the posterior mean (which is indeed crucial), but GPs are often valued for their uncertainty estimates. The quality of uncertainty quantification isn't clearly demonstrated here.

2) There are some motivational gaps, which I address in the “Questions” section. I assigned a significance score of 2 because the problem being addressed feels quite specific to the GP community. I'm open to being convinced otherwise.

3) Some aspects of the solution are not explained in sufficient detail (e.g., the notion of "condition number-free"), which affected my clarity and confidence scores.

---

> ### Author Rebuttal · Authors · 2025-07-30
>
> > **Comment 1.1** — The justification for the algorithm primarily focuses on the posterior mean
> (which is indeed crucial), but GPs are often valued for their uncertainty estimates. The quality of
> uncertainty quantification isn’t clearly demonstrated here.
>
> Our method can be used for both mean and uncertainty estimates, as demonstrated in our
> Bayesian optimization experiments, which require both. Our exposition focuses on the mean just for
> clarity. We'll clarify in the paper that our method can handle both the mean and uncertainty estimates.
>
> > **Comment 1.2** — When the dataset becomes very large (as in the main scenario considered in
> the paper), in my experience, it often makes sense to move away from GPs and use alternative
> models like Bayesian Neural Networks or Deep Kernel Methods, which don’t suffer from the same
> scaling issues. Could the authors elaborate on why GPs remain a preferable choice in this context?
> Some comparative discussion would strengthen the case for GPs.
>
> While Bayesian neural networks (BNNs) and deep kernel methods (DKMs) are attractive
> alternatives for Bayesian modeling, we believe that GPs still retain advantages in certain scenarios.
>
> First, regarding scalability, current state-of-the-art implementations of BNNs and DKMs rarely scale
> to large datasets (e.g., $n \gg 10^6$ samples) (Ober et al., 2021; Arbel et al., 2023). In contrast, our work
> demonstrates full GP inference on datasets with $n \sim 10^8$ samples, which is two orders of magnitude
> larger than what is typically feasible for BNNs or DKMs.
>
> Second, in terms of prior knowledge and interpretability, GPs allow practitioners to specify kernels
> informed by domain expertise. If a suitable kernel is already known, using GPs would be advantageous
> over learning the kernel via DKMs. Furthermore, DKMs are prone to overfitting: Ober et al. (2021)
> shows that DKMs can overfit even more than neural networks trained using the standard maximum
> likelihood approach.
>
> > **Comment 1.3** — Existing state-of-the-art GP methods already handle datasets on the order
> of $10^6$. So, could authors provide any details on how much additional value is gained by pushing
> beyond this scale? This ties into the previous question about the motivation for continuing with
> GPs at such scales.
>
> We live in an era of big data, where datasets often have size  $n > 10^{6}$.
> For datasets that are this large, methods like PCG are unsuitable for GP inference (e.g., taxi, for which $n \geq 10^{8}$).
> Methods like SDD can scale to datasets of this size, but they have to be carefully tuned to achieve optimal performance, as Figure 3 shows.
> However, such tuning is prohibitively expensive (see the runtimes in Figure 3).
> Moreover, even for datasets where $n \sim 10^{6}$, such as houseelec in Figure 1, existing SOTA solvers cannot match the performance of ADASAP.
> For PCG, this arises because of the time required to complete each iteration, while SDD performs poorly (even when tuned) due to ill-conditioning of $K + \lambda I$.
> Thus, in addition to being essential for tackling problems of larger scale, ADASAP solves problems at previous limit scales much more efficiently than existing SOTA solvers when $K + \lambda I$ is ill-conditioned.
> This is crucial as kernel matrices are often ill-conditioned in practice.
>
> > **Comment 1.4** — I didn’t fully grasp two points that might be critical. Could the authors clarify what they mean by “condition number-free” and what exactly is considered “ill-conditioning”? I assume this refers to challenges in inverting the matrix $K + \lambda I$, but I’m not entirely certain that’s the intended meaning.
>
> We are happy to clarify our terminology.
> The reviewer is correct that ill-conditioning refers to the conditioning of the kernel matrix $K + \lambda I$.
> $K + \lambda I$ is ill-conditioned when the condition number $(\lambda_1(K)+\lambda)/(\lambda_n(K)+\lambda)$ is large.
> As discussed in Section 2.2 (Line 84), the convergence rate of SDD, a SOTA exact inference method, is controlled by the condition number of $K + \lambda I$.
> Thus, SDD converges slowly on problems where $K +\lambda I$ is ill-conditioned, as can be seen in our experiments.
> Corollary 3.3 (Line 169) shows that for certain levels of precision, the convergence rate of the posterior mean along the top $\ell$-basis functions is completely independent of the condition number of $K + \lambda I$ and the value of $\lambda$.
> See Remark 3.4 (Line 172) for details.
> This is what we mean when we say ``condition number-free'' in the paper.
> We will make the meaning of this terminology more explicit in the revision.
>
> > **Comment 1.5** — In Algorithm 1, the variable ‘b’ appears without an immediate explanation. While it becomes clearer later in Section 3.2, it would help to define it properly when it first appears, especially since it plays a role in justifying the SAP algorithm’s reduced cost. Also, practical guidance on how to choose the block size ‘b’ would be very helpful.
>
> Thanks for pointing this out.
> We agree that the blocksize $b$ should be defined earlier in the paper---we'll fix this in the revision.
>
> > **Comment 1.6** — Could the authors provide references or direct links to the proofs of Theorem 3.2 and Corollary 3.3 in the Appendix when they are first introduced? Currently, it’s a bit inconvenient to search the paper and then the Appendix for these proofs.
>
> We will add the references to the proofs of Theorem 3.2 and Corollary 3.3 to the main paper.
>
> > **Comment 1.7** — However, the discussion of limitations and potential future work could be expanded. For example, what kinds of problems are particularly challenging for the method? How do the authors recommend practitioners use the method in different contexts? It would be helpful to draw comparisons to non-GP alternatives.
>
> The most challenging type of problem for ADASAP is one in which the posterior mean is uniformly spread across the spectral basis functions of the RKHS $\mathcal H$.
> In this case, learning the posterior mean well along the top spectral basis functions is insufficient for achieving good predictive performance.
> We note this challenge is not unique to ADASAP, since methods like PCG and approximate kernel methods also suffer in this regime.
> Fortunately, this setting is atypical for kernel and GP problems.
> As we mention in the beginning of Section 3, many prior works (Dicker et al., 2017; Lin et al., 2020;
> 2023) have shown that the top $\ell$-spectral basis functions (for some $\ell \ll n$) are most responsible for achieving good generalization both in theory and practice.
> Nevertheless, we will add discussion of this problem class to the revision.
>
> We believe ADASAP will be valuable to practitioners whenever they are using GP and Bayesian methods that require the solution of large kernel linear systems.
> We would recommend replacing solvers like Choleksy and PCG (which are unfriendly for large-scale problems) with ADASAP.
> Note that linear systems of this size arise in models that go beyond the basic GP setting.
> For instance, we believe ADASAP could be valuable for scaling deep kernel earning to larger datasets, as certain derivative calculations in the log-likelihood require solving a linear system with a large kernel matrix (see Eq. 7 in Wilson et al. (2016)).
> We will include this discussion in our updated revision.
>
> > **Comment 1.8** — While the authors emphasize that the algorithm performs well with default hyperparameters (which is great for adoption), more discussion around what might go wrong and how one should adjust the hyperparameters accordingly would be useful for practitioners.
>
> We agree that the submission could benefit from including more details about how to tune the hyperparameters for ADASAP. Here are some examples, for intuition:
> * Blocksize $b$: Increasing $b$ should lead to faster per-iteration convergence, at the cost of more expensive iterations (both in terms of wall-clock time and memory).
> On the other hand, decreasing $b$ should lead to slower per-iteration convergence, while leading to cheaper iterations.
> In general, we recommend selecting $b$ to maximize GPU utilization without running out of memory.
>
> * Rank $r$: The behavior for adjusting $r$ is similar to that of adjusting $b$.
> Larger values of $r$ lead to faster convergence, but more expensive iterations; smaller values of $r$ lead to slower convergence, but cheaper iterations.
>
> * Acceleration parameters $\mu$ and $\nu$: Increasing $\mu$ (or decreasing $\nu$) could improve the convergence rate in practice.
> However, practitioners should be careful to ensure that $\mu < \nu$, otherwise this will make some of the Nesterov acceleration parameters negative, which could lead to unstable behavior/divergence.
>
> **References**
>
> Julyan Arbel, Konstantinos Pitas, Mariia Vladimirova, and Vincent Fortuin. A primer on bayesian
> neural networks: Review and debates. arXiv preprint arXiv:2309.16314, 2023.
>
> Lee H Dicker, Dean P Foster, and Daniel Hsu. Kernel ridge vs. principal component regression:
> Minimax bounds and the qualification of regularization operators. Electronic Journal of Statistics,
> 11:1022–1047, 2017.
>
> Jihao Andreas Lin, Javier Antoran, Shreyas Padhy, David Janz, José Miguel Hernández-Lobato,
> and Alexander Terenin. Sampling from Gaussian process posteriors using stochastic gradient
> descent. In Advances in Neural Information Processing Systems, 2023.
>
> Junhong Lin, Alessandro Rudi, Lorenzo Rosasco, and Volkan Cevher. Optimal rates for spectral algorithms with least-squares regression over Hilbert spaces. Applied and Computational Harmonic
> Analysis, 48(3):868–890, 2020.
>
> Sebastian W Ober, Carl E Rasmussen, and Mark van der Wilk. The promises and pitfalls of deep
> kernel learning. In Uncertainty in Artificial Intelligence, pages 1206–1216, 2021.

---

> > ### Comment · Reviewer_P6N1 · 2025-08-01
> >
> > Thank you, I appreciate the authors’ response and the clarifications provided. I have found them really useful.
> >
> > However, I still believe that the quality of the uncertainty estimation would ideally warrant further evaluation. In particular, additional testing and practical comparisons with alternative methods could significantly strengthen the paper.
> >
> > As such, my overall assessment remains unchanged.

---

### Decision · Program_Chairs · 2025-09-17

**Decision:**

Accept (poster)

**Comment:**

The reviewer consensus is that this is a good paper. Reviewers praise high performance, theoretical grounding in terms of convergence results, and the ability to implement in a distributed manner as strengths. Reviewers also praise clarity, and describe the work as well-written. Most of the weaknesses have to do with minor typos, and some presentation details with how ill-conditioning is discussed (Reviewer WCDC, who gave the most critical score). From the text of the reviews, I infer that the reviewers see the strengths as significantly outweighing the weaknesses.

The reviewer consensus is that it should be accepted, and I concur.